# How to Keep Investors' Confidence after Being Labeled as Polluting Firms: The Role of External Political Ties and Internal Green Innovation Capabilities

Liangdong Lu, Mengyao Wang and Jia Xu *

School of Business, Hohai University, Nanjing 210098, China; luliangdong@hhu.edu.cn (L.L.)
* Correspondence: jiaxu@hhu.edu.cn

**Abstract:** This study delves into investors' perceptions of the polluting label attached to listed manufacturing firms, emphasizing the interplay between external political ties and internal green innovative capability in influencing these perceptions. Drawing on a longitudinal analysis of listed manufacturing firms in China from 2010 to 2020 and employing a difference-in-differences (DID) approach, we treat firms identified under the National Specially Monitored (NSM) program as the treated group, while non-NSM firms form the control group. The time variable captures the period post the introduction of the NSM program. Our findings highlight that the polluting label created a loss prospect for investors, signifying diminishing returns over time. Interestingly, firms with closer connections to local governments experienced amplified negative investor perceptions. In contrast, strong affiliations with the central government and robust green innovative capabilities cushioned these adverse reactions. Notably, central ties proved even more beneficial when complemented by green innovative capability. By melding signal theory with the literature on sense-making, this research adds nuance to the discourse on the role of resources in determining firm success amidst environmental controversies.

**Keywords:** polluting label; political ties; innovative capability; sense-making; market value

## 1. Introduction

The challenges of sustainable development are increasing, especially for manufacturing firms facing rigorous environmental management institutional pressures [1]. Despite the extensive body of research focused on how organizations address these challenges, there remains a gap in understanding the perceptions of external stakeholders—primarily investors—towards these pressures. Investors operate in a complex environment filled with uncertainty. In this backdrop, they continually strive to make sense of unforeseen events and determine how to respond to them [2].

What sets our research apart is its exploration of the interplay between investor perceptions and organizational characteristics, a dimension less traversed in literature. Understanding investor behavior becomes pivotal, as it holds the potential to redefine how firms prioritize and implement sustainability practices. This study emanates from a pressing need to decipher investor reactions when their invested firms confront the duality of institutional pressures, particularly when labeled as polluting entities. This research is distinguished by its commitment to not only decode investor reactions but also probe the influence of certain firm traits, such as political affiliations and eco-innovation competencies, in sculpting these reactions. With this in mind, this study sets forth with the following primary objectives:

First, to ascertain how investors interpret the polluting label of listed manufacturing firms.

Second, to understand the perception of a firm's political ties and green innovative capabilities by investors, especially in the context of the polluting label.

Third, to uncover how these political ties and innovation capabilities, when combined, mold investors' understanding of the polluting label.

To provide empirical weight to our objectives, this study critically analyzes the market value of listed manufacturing firms in China from 2010 to 2020. Our focus is on the National Specially Monitored (NSM) program, launched in 2007 to oversee heavy polluting firms. Given its nature, the NSM program serves as a distinctive quasi-experimental platform that facilitates causal inference. Moreover, with China's impending mandate for environmental data disclosure for its publicly listed firms, our exploration finds relevance by shedding light on the underlying reasons driving this transition towards greater environmental responsibility.

The role of investor perceptions in determining corporate actions cannot be understated. Recent studies have indicated that investors are increasingly considering environmental, social, and governance factors in their investment decisions, further underlining the importance of corporate environmental responsibility [3]. Moreover, in emerging markets such as China, where environmental degradation has been a significant concern, investors are uniquely positioned to drive corporate sustainability practices through their capital allocation decisions [4].

Furthermore, the association between a firm's political ties and its market valuation, particularly in the context of environmental controversies, has been the focus of recent investigations. Wu et al. (2018) noted that while political ties can offer protection and preferential resources in some contexts, they might also bind firms to governmental agendas and lead to entrenchment risks [5]. On the other hand, a firm's green innovation capabilities are emerging as a prominent factor in enhancing market value. Green innovation, as Huang and Li (2017) highlighted, can act as a significant differentiator in investor perceptions, offering firms a competitive advantage in a market that is increasingly prioritizing sustainable practices [6].

China's unique institutional environment, with its interplay between government, industry, and market forces, offers a fertile ground for such investigations. The Chinese government's rigorous push towards a more sustainable and green economy, as evidenced by its Five-Year Plans and other policy documents, further intensifies the need for a deeper understanding of investor behavior in the country's listed manufacturing sector [7].

With these perspectives, this study contributes to the growing body of literature that seeks to bridge the understanding of investor perceptions, corporate political ties, green innovation capabilities, and their combined influence on firm valuation in the face of environmental controversies.

## 2. Theoretical Framework and Hypothesis Development

### 2.1. Theoretical Roadmap: Goal, Hypotheses, and Methods

Table 1 presents a structured approach detailing our research objectives, the corresponding hypotheses, and the methods that will be employed to test each hypothesis:

### 2.2. Investor's Perception of Polluting Label

Investors evaluate a firm's environmental commitment and overall quality when confronted with a polluting label, assessing its corporate resources, capabilities, and potential for sustainable growth. This evaluation fundamentally hinges on prospect theory [8], which postulates that individuals assess decisions predicated on anticipated gains or losses.

The nature of these gains and losses, and the resulting risk attitudes, have been the focus of extensive research. For instance, Van Dijk and Van Knippenberg (1996) found that heightened uncertainty can exacerbate loss aversion [9]. This is further corroborated by Shogren et al. (1994), who illustrated the intensified sensitivity towards irreplaceable losses [10]. Additionally, the influence of acquisition difficulty on the judgment of gains and losses was underlined by Loewenstein & Issacharoff (1994) [11].

**Table 1.** Roadmap of this study.

| |
| --- |
| **Objective 1:** Examine investors' perception of polluting labels and how this perception changes over time. |
| - Hypothesis 1: The polluting label evokes negative perceptions among investors, but its impact lessens as time progresses.<br>- Method: DID analysis. |
| **Objective 2:** Understand the perception of a firm's political ties and green innovative capabilities by investors, especially in the context of the polluting label. |
| - Hypothesis 2a: Ties with the central government positively influence investor perceptions.<br>- Hypothesis 2b: Local political ties might be perceived negatively by investors.<br>- Hypothesis 3: A strong green innovative capability is likely to foster positive investor perceptions.<br>- Method: Two-stage approach. |
| **Objective 3:** Uncover how these political ties and innovation capabilities, when combined, mold investors' understanding of the polluting label. |
| - Hypothesis 4a: The prominence of central ties positively moderates the negative impact of local ties on investors' perception.<br>- Hypothesis 4b_1 and Hypothesis 4b_2: Green innovative capability has an influence on perceptions of local ties.<br>- Hypothesis 4c: The positive perception of innovative capability by investors is overshadowed by prominent central ties.<br>- Method: Interaction effect test. |

The ever-evolving discourse on sustainability and corporate responsibility has deepened the complexities surrounding the polluting label. Firms that appear on lists such as the NSM often send out signals of prioritizing immediate profits over sustainable practices. This, in turn, places them under the scrutiny of both public and regulatory bodies. In the face of these signals, it is plausible to assume that investors' loss perceptions are heightened, given the potential reputational and financial implications for the firm.

In the modern context, research has emphasized the financial implications of environmental disregard. For instance, one study highlighted that carbon-intensive companies, or those flagged for high $CO_2$ emissions, often experience a stock market penalty, reinforcing the financial repercussions of the polluting label [12,13]. Another notable study elucidated that, increasingly, investors, especially institutional ones, are integrating Environmental, Social, and Governance (ESG) criteria into their investment decisions [14]. This implies that labels like "polluting" can indeed have a profound impact on investors' perceptions and decision-making processes.

However, prospect theory posits that individuals might demonstrate heightened risk tolerance in the face of negative outcomes. Given this, it is conceivable that investors might retain holdings in firms with long-standing presence on the NSM list, banking on the potential for future rebounds or strategic shifts towards sustainability. Over time, this might attenuate the initially potent negativity associated with the polluting label.

**Hypothesis 1 (H1):** *The polluting label evokes negative perceptions among investors, but its impact lessens as time progresses.*

### *2.3. Roles of Political Ties and Green Innovative Capability on Investors' Perception*
2.3.1. Political Ties

Firms' strategic relationships with central governments can confer considerable advantages, and this has been extensively documented in academic literature [15]. The nature and potential of these ties traverse beyond mere information access and can profoundly shape a firm's trajectory. An empirical examination from China illuminates how such political affiliations considerably bolster a firm's innovation capabilities [16]. This sheds light on the multifaceted nature of these ties, suggesting they also act as catalysts that spur innovation by providing firms with advantageous resources and support.

These benefits of central government ties also encompass enabling firms to gain a deeper understanding of policy design, anticipate policy shifts, and adeptly navigate the implications of those shifts [17,18]. Such foresight and adaptability are invaluable, especially given the dynamic and often unpredictable nature of the political landscape.

Firms with central government ties also enjoy reduced uncertainty, especially in environments characterized by volatile regulatory frameworks [19]. By tapping into these relationships, firms access privileged insights, positioning them in a proactive rather than a reactive stance [20]. Yet, while these ties confer numerous advantages, they also entail specific responsibilities and expectations. Companies that deviate from central government guidelines, or fail to capitalize on their relationships effectively, can face severe repercussions, ranging from reduced support to punitive actions [21,22].

Incorporating the above understanding, it becomes evident that the presence and depth of ties with the central government can considerably sway investor perceptions. Investors, recognizing the multifaceted advantages of such ties, are likely to view them as indicators of a firm's strategic positioning and potential resilience against policy shocks.

**Hypothesis 2a (H2a):** *Ties with the central government positively influence investor perceptions.*

Local political ties, epitomized by affiliations at subnational levels such as provinces, can serve as a double-edged sword for firms [23]. On the one hand, these ties can be instrumental in reducing regulatory uncertainty, facilitating smoother access to local resources, and potentially aiding in swift navigation of regional bureaucracies [24]. Furthermore, they can engender a closer alignment with local stakeholders, boosting a firm's social capital and reputation within the community [25].

However, challenges arise when there is a misalignment between local and central objectives. Such divergences can place firms in precarious situations. While local ties might help navigate regional challenges, they might simultaneously complicate a firm's relationship with central authorities, especially when local agendas clash with broader national mandates. Firms heavily embedded in local politics might also face challenges if regional leadership undergoes significant change or if there is a major policy shift at the local level [26]. Consequently, the potential risks associated with these ties could overshadow their benefits, especially in the eyes of investors who are keenly aware of the balancing act firms must perform between local affiliations and central directives [27,28].

Given the nuanced interplay of benefits and potential detriments associated with local political ties, investors' perceptions might lean towards caution. This cautious outlook stems from the inherent risks these ties introduce, especially when considering the overarching objectives of programs like the NSM and their potential repercussions.

**Hypothesis 2b (H2b):** *Local political ties might be perceived negatively by investors.*

2.3.2. Green Innovative Capability

The realm of green innovation has received increased attention as both an environmental imperative and a business opportunity. Delving deeper than the earlier observation of the importance of sustainability-oriented dynamic capabilities, recent research underscores its role in positioning firms strategically in an eco-conscious market [29]. Such sustainable strategies, as corroborated by Hart and Dowell (2011), can provide dual benefits—they not only cater to growing global environmental concerns but also drive a competitive edge in the market [30].

A key dimension of green innovation is its capacity to synergize economic performance with environmental sustainability. This encompasses the development of new products, practices, and organizational processes that simultaneously boost economic returns and reduce ecological footprints [31]. In the evolving landscape, green innovation is not just an add-on; it is becoming integral for companies aiming for long-term success. Notably, firms with a polluting label stand to gain considerably by demonstrating a robust commitment to green innovation, as this can potentially offset negative perceptions and highlight a proactive approach to sustainability [32].

Moreover, Chen et al. (2019) emphasize the importance of integrating green innovation strategies within core business models. They argue that such integration not only

helps firms meet stringent regulatory standards but also positions them favorably among environmentally-conscious investors [33]. Thus, in the backdrop of these insights, and considering the criticality of green innovative capabilities for labeled polluters, the following hypothesis is presented:

**Hypothesis 3 (H3):** *A strong green innovative capability is likely to foster positive investor perceptions.*

*2.4. Interplay between Political Ties and Innovative Capability in Influencing Investors' Perception*

2.4.1. Interaction of Central and Local Political Ties

Firms often navigate the political landscape by fostering relationships at both the central and local levels. While the intertwined nature of these relationships offers a layered understanding, the coexistence may sometimes lead to investor apprehension [34]. A firm's affiliation with the central government tends to offer advanced access to resources, policy insights, and a reduced risk of punitive actions [17]. Drawing upon the certainty effect embedded in prospect theory, investors are likely to gravitate towards the tangible benefits emanating from central political ties, potentially overshadowing the intricacies and perceived risks associated with local affiliations.

**Hypothesis 4a (H4a):** *Central political ties serve as a buffer, attenuating the potential adverse impact of local ties on investors' perception.*

2.4.2. Interaction of Local Ties and Innovative Capability

Recent studies have illuminated the intricacies of innovation within the context of knowledge sharing and transfer, highlighting both potential roadblocks and catalysts [35]. When local political ties enter the equation, they can either bolster or hinder a firm's innovative pursuits, contingent on the nature of these relationships. With the continuing debate on whether relationships and capabilities offer alternative or complementary value propositions [36], understanding the investor perception becomes crucial. Drawing from both prospect theory and portfolio selection theory [37], the intricate balance of green innovation's inherent risks and the uncertainties underpinned by dominant local ties can result in differential investor sentiments.

**Hypothesis 4b_1 (H4b_1):** *A robust green innovative capability acts as a counterbalance, offsetting the negative investor perceptions tied to local political affiliations.*

**Hypothesis 4b_2 (H4b_2):** *Intense green innovative capability, when juxtaposed with prominent local ties, exacerbates investor concerns due to compounded uncertainties.*

2.4.3. Interaction of Central Ties and Innovative Capability

For firms deeply entrenched with the central government, the interplay between political ties and innovative capabilities offers a unique dimension. While innovation can herald adaptability, growth, and potential label mitigation, the journey is riddled with uncertainties [38,39]. In contrast, central political affiliations can serve as a beacon of stability, often translating to economic rewards and a competitive edge [40,41]. Drawing from the irreplaceability principle of Shogren et al. (1994) and the recent findings of Wang et al. (2021), which indicate that investors often prefer known certainties over unknown risks [10,42], it can be posited that robust central affiliations might eclipse the perceived benefits of innovation.

**Hypothesis 4c (H4c):** *The positive perception of innovative capability by investors is overshadowed by prominent central ties.*

## 3. Methodology

### 3.1. Sample and Data Collection

The goal of this study is to examine investors' reactions when their focal firms confronted incompatible institutional pressures. Therefore, our sample consists of all manufacturing firms that were publicly listed on the Shenzhen and Shanghai Stock Exchanges from 2010 to 2020. This study employs multiple sources to construct our dataset. China's NSM policy provides a perfect background for us to test our research framework, as NSM firms are typically marked by the central government as polluters to be monitored. The list of NSM firms is collected from the website of the Ministry of Environmental Protection (MEP) of China. Table 2 shows the statistics of the number of firms, the number of listed manufacturing firms, and the corresponding monitoring time included in the annual list. By matching the name and address of the enterprise, the included listed manufacturing firms can be found. The sample selection rules that are adopted for the different hypotheses are shown in Table 3. Other firm-level information mainly comes from the China Stock Market and Accounting Research (CSMAR) database and IncoPat global patent database (IncoPat fully collects more than 100 million basic patent data from 112 countries/organizations/regions (including 22 major countries) around the world, providing bilingual retrieval of global patents in both Chinese and English). For province-level data, statistics yearbooks published by the National Bureau of Statistics of China and the China Statistical Yearbook on Environment are used.

**Table 2.** Annual distribution of NSM firms.

| Year | Number of Listed Manufacturing Firms Newly on the List | Monitor Duration (Years) | | | | | | | | | Total Number of Listed Manufacturing Firms in That Year (Excluding Firms That Have Stopped Production) | Sample Size after 1:1 PSM Matching Year by Year |
|------|------|-----|-----|-----|-----|-----|-----|-----|-----|-----|------|------|
| | | 1 | 2 | 3 | 4 | 5 | 6 | 7 | 8 | 9 | | |
| 2010 | 114 | | | | | | | | | | 114 | 44 |
| 2011 | 143 | 81 | | | | | | | | | 224 | 81 |
| 2012 | 142 | 95 | 43 | | | | | | | | 280 | 104 |
| 2013 | 131 | 78 | 51 | 19 | | | | | | | 279 | 112 |
| 2014 | 155 | 83 | 49 | 11 | 12 | | | | | | 310 | 172 |
| 2015 | 160 | 74 | 46 | 28 | 18 | 11 | | | | | 337 | 209 |
| 2016 | 183 | 65 | 36 | 21 | 20 | 17 | 8 | | | | 350 | 201 |
| 2017 | 276 | 97 | 47 | 22 | 17 | 5 | 9 | 4 | | | 477 | 232 |
| 2018 | 253 | 123 | 80 | 36 | 15 | 7 | 15 | 5 | 4 | | 538 | 239 |
| 2019 | 154 | 114 | 49 | 53 | 12 | 2 | 5 | 6 | 2 | | 397 | 186 |
| 2020 | 24 | 25 | 13 | 5 | 22 | 3 | 1 | 5 | 5 | 2 | 105 | 90 |
| Total | 1735 | 835 | 414 | 195 | 116 | 45 | 38 | 20 | 11 | 2 | 3411 | 1670 |

Note: Manufacturing firms refer to the listed firms in the industries C13–C42 under the China Securities Regulatory Commission classification system.

**Table 3.** Sample selection rules adopted for different hypotheses.

| Hypothesis | Sample Description |
|------------|--------------------|
| H1 | All listed manufacturing firms (i.e., Industries C13–C42 under the China Securities Regulatory Commission classification system) on the NSM list. The total amount of firms is 269. |
| H2,3 | According to the research design, the PSM processing group in this paper were manufacturing firms listed as NSM firms from 2010 to 2020. The control group were listed manufacturing firms that had never been listed as NSM firms from 2010 to 2020. Then, the probit model was used to estimate the propensity score, kernel matching was used to determine the weight, and common support conditions were applied for matching year by year. The final number of observations is 1580, including 1670 NSM firms and 1670 non-NSM firms. |
| H4 | 1670 observations that are NSM-listed manufacturing firms from 2010 to 2020. |

Note: (1) The control variables adopted for matching are: firm age, CSR score, environmental malpractice disclosure, leverage, institutional ownership, shareholder concentration (CRIO), shareholder concentration (Z index), ROA, firm size, report attention, transparency, environmental subsidies, development capability, organizational slack, prior R&D expenditure, prior political ties, TMT age heterogeneity, TMT education heterogeneity, TMT tenure heterogeneity, TMT occupation heterogeneity, CEO duality, stakeholder attention, Tobin's Q, local formal regulation, local informal regulation, local pollution, local government fiscal power, subindustry (dummy), and type of actual controller (dummy). (2) In addition, due to the lack of data for many key control variables (CSR score, environmental malpractice disclosure, etc.) before 2010, the 2009 samples were excluded.

Given the NSM policy context, there is a requirement to compare firms labeled as polluters with those not labeled as such. The PSM-DID approach enables us to match these

firms more effectively, ensuring they are comparable prior to the intervention, allowing for a more accurate estimation of the policy's effect on investor perceptions. PSM assists in reducing treatment selection bias, ensuring that the treated and control groups are balanced on observed pretreatment covariates. The DID strategy aids in controlling for unobserved heterogeneities that remain constant over time, capturing the precise effect of the NSM policy on investor perceptions.

### 3.2. Measurement

*Dependent variable.* To create our dependent variable, buy-and-hold abnormal return (BHAR) is employed to measure long-term abnormal returns [43–45]. The main concern of this study is to compare the actual return of the listed firm's stock with the expected return after the release of the NSM list. After calculating the excess return, it is used to measure whether the firm had been recognized by the market. BHAR measures the excess return on a firm's stock over the return on its market portfolio by buying and holding it until the end of the study period [43]. It closely mimics the shareholder experience, in the form of risk-adjusted returns over a holding period [44]. Its characteristic-based matching portfolio approach assumes an investment-oriented buy-and-hold strategy for the firm, confronting the NSM polluting label by holding the stock for a whole year while the firm is on the list according to the BHAR calculation. Since the NSM program regularly publishes the list of firms that need to be monitored in the next year on December 31 of each year, this study calculates the BHAR value for 250 trading days starting on the day of announcement. The calculation formula of BHAR is as follows:

$$BHAR_{it} = \prod_{t=0}^{T}(1+R_{it}) - \prod_{t=0}^{T}[1+E(R_{it})] \qquad (1)$$

where $T$ represents the time interval of the inspection, $T = 0\sim250$, $t = 0$ represents the day the list is published, $t = 1$ represents the first day after the list is published, and so on. $R_{it}$ is the stock return rate of the sample firms on $t$ trading day and $E(R_{it})$ represents the expected return on the stock of the sample firms on $t$ trading day.

### 3.3. Independent Variables

*Political ties.* Prior studies tended to examine whether a firm's political connection would affect them [46]. Plummer et al. (2016) used continuous variables rather than binary variables [47]. Similarly, this study represents the degree of political ties as a continuous variable, by calculating the cumulative value of the administrative level of the executives' positions in government agencies. The formulas for central and local political ties were as follows, respectively:

$$\text{Central political ties} = \sum_{i=1}^{n} C_i \qquad (2)$$

where i = (1, n), n is the number of top management team (TMT) executives who served as delegates to central institutions, and $C_i$ is the administrative level of the executive's position: 5 = national level; 4 = provincial level; 3 = bureau level; 2 = county level; and 1 = township and other levels.

$$\text{Local political ties} = \sum_{i=1}^{n} L_i \qquad (3)$$

where i = (1, n), n is the number of TMT executives who served as delegates to local institutions (all institutions at the provincial or ministerial level, departmental level, county level, township level, and so on), and $L_i$ is the administrative level of the executive's position: 5 = national level; 4 = provincial level; 3 = bureau level; 2 = county level; and 1 = township and other levels.

*Green innovative capability.* Patent data come from the IncoPat global patent database. In 2010, the World Intellectual Property Organization (WIPO) launched an online tool that was designed to facilitate the retrieval of patent information related to environmentally

friendly technologies, called the Green List of International Patent Classification. It classifies seven major green patents based on the United Nations Framework Convention on Climate Change, including transportation, waste management, energy conservation, alternative energy production, administrative regulatory or design aspects, agriculture or forestry, and nuclear power generation. According to the above classification criteria, this study identifies and calculates the number of enterprises' green patents every year, and further distinguishes the green invention patents and green utility patents as the core measurement indicators of listed manufacturing firms.

### 3.4. Control Variables

This study includes the following variables that were used in prior studies to control for investors' reaction [48–51]: firm age, comprehensive ranking system (CSR) score, environmental malpractice disclosure, leverage, institutional ownership, shareholder concentration (CRIO), shareholder concentration (Z index), return on assets, firm size, report attention, transparency, environmental subsidies, development capability, organizational slack, prior research and development (R&D) expenditure, prior political ties, TMT age heterogeneity, TMT education heterogeneity, TMT tenure heterogeneity, TMT occupation heterogeneity, chief executive officer duality, stakeholder attention, Tobin's Q, local formal regulation, local informal regulation, local pollution, local government fiscal power, subindustry (dummy), and type of actual controller (dummy). The detailed measurements and sources of each variable are shown in Table 4.

**Table 4.** Measurement of variables.

| Variable name | Measurement |
|---|---|
| 1. Market value | BHAR. |
| 2. Polluting label (dummy) | 1 = focal manufacturing firm is NSM firm; 0 = otherwise, annual. |
| 3. Central tie | Central political tie $= \sum_{i=1}^{n} I_i$ where i = (1, n), n is the number of TMT executives who served as delegates to central institutions; $A_i$ is the administrative level of the executive's position: 5 = national level; 4 = provincial level; 3 = bureau level; 2 = county level; and 1 = township and other levels. |
| 4. Local tie | Local political tie $= \sum_{i=1}^{n} I_i$ where i = (1, n), n is the number of TMT executives who served as a delegate to local institutions; $I$ is the administrative level of the executive's position: 5 = national level; 4 = provincial level; 3 = bureau level; 2 = county level; and 1 = township and other levels. |
| 5. Green innovative capability | Log(number of green invention patents + number of green utility patents + 1). |
| 6. Firm age | Log[(year that the firm was monitored by MEP − year that the firm was founded), 10]. |
| 7. CSR score | Log(CSR grade disclosed by HEXUN in that year + 1, 10), annual. |
| 8. Environmental Malpractice disclosure | The number of times the firm was disclosed by local environmental protection bureaus (EPBs) or MEP to have violated environmental regulations. |
| 9. Leverage | Debt/Assets, annual. |
| 10. Institutional ownership | Percentage of shares owned by all institutional owners, annual. |
| 11. Shareholder concentration (CRIO) | Percentage of shares owned by the top ten largest shareholders, annual. |
| 12. Shareholder concentration (Z index) | Percentage of shares owned by the largest shareholder/percentage of shares owned by the second largest shareholder, annual. |
| 13. ROA | Return on assets in that year, annual. |
| 14. Firm size | Log(total assets, 10), annual. |
| 15. Report Attention | Log(number of research reports that have tracked and analyzed the focal firm + 1,10), annual. |
| 16. Transparency | Log(the total number of announcements published by the focal firm, 10), annual. |
| 17. Environmental subsidies | Variable was retrieved and calculated from tables specifying "details for subsidies from the government," which are provided in the firm's annual reports. |
| 18. Development capability | Revenue growth rate in that year, annual. |
| 19. Organizational slack | Log(major repair fund, 10) + log(inventory fund, 10) + log(accounts payables, 10), annual. |
| 20. TMT Age Heterogeneity | Age standard deviation/mean age, annual. |

**Table 4.** *Cont.*

| Variable name | Measurement |
|---|---|
| 21. TMT Education Heterogeneity | TMT Education Heterogeneity $= 1 - \sum P_i(LnP_i)$ where i = (1, N), N is the number of academic qualifications. This paper examines five types of educational background: doctor or above, master's degree, bachelor's degree, junior college degree, high school degree or below; $P_i$ represents the percentage of the executive team with a certain degree, annual. |
| 22. TMT Tenure Heterogeneity | Tenure standard deviation/mean tenure, annual. |
| 23. TMT Occupation Heterogeneity | TMT Occupation Heterogeneity $= 1 - \sum P_i(LnP_i)$ where i = (1, N), N is the number of functional experience categories. This paper examines nine types of career background: production, research and development, design, human resources, management, marketing, finance, finance, and law. $P_i$ represents the percentage of the executive team with experience in a specific function, annual. |
| 24. CEO duality | 1 = CEO is also the chairman of the board; 0 = otherwise, annual. |
| 25. Stakeholder Attention | Stock turnover in that year, annual. |
| 26. Tobin's Q | Tobin's Q in that year, annual. |
| 27. Local formal regulation | Entropy-weighted average of number of firms punished for environmental pollution, total revenue of sewage charge fees, number of public complaints completed, and number of laws and regulations, annual. |
| 28. Local informal regulation | Entropy-weighted average of number of environmental protection initiatives submitted by the National People's Congress, number of environmental protection initiatives submitted by the Chinese People's Political Consultative Conference, and number of public environmental complaints, annual. |
| 29. Local pollution | Entropy-weighted average of total volume of wastewater discharged, total volume of COD discharged, total volume of ammonia nitrogen discharged, total volume of sulfur dioxide discharged, total volume of soot and dust discharged, and total volume of nitrogen oxide discharged, annual. |
| 30. Local government fiscal power | Log(total provincial government revenue,10), annual. |

### 3.5. Model and Analysis

*Analysis of investors' sense-making of the polluting label.* Drawing on Beck et al. (2010)'s approach [52], this study employs a dynamic model to analyze the impact of the polluting label on investor perception and its changes over time. The constructed dynamic model is as follows:

$$BHAR_{st} = \alpha + \beta_1 D_{st}^{-5} + \beta_2 D_{st}^{-4} + \cdots + \beta_5 D_{st}^{-1} + \beta_6 D_{st}^{1} + \beta_7 D_{st}^{2} \cdots + \beta_{15} D_{st}^{10} + A_S + B_t + \varepsilon_{st} \quad (4)$$

where $Ds$ equal zero, except as follows: $D^{-i} = 1$ for firms in the $i_{th}$ year before the introduction of the NSM program, while $D^i$ equals one for firms in the $i_{th}$ year on the list. $A_S$ and $B_t$ are vectors of the NSM firm and year dummy variables that account for firm and year fixed effects, respectively.

*Analysis of the effects of political ties and innovative capability on investors' sense-making of polluting label.* Given that political ties and innovative capability might be correlated with other unobserved variables, this can introduce estimation bias. The two-stage approach allows us to consider all known control variables initially, and then in the second stage estimate our primary explanatory variables using the residuals from these controls. This procedure mitigates potential endogeneity issues. A two-stage model was developed to examine the effects of political ties and innovative capability on the investors' reactions to all 1580 firms obtained by a 1:1 year-by-year propensity score matching (PSM), including 1670 NSM and 1670 non-NSM manufacturing firms. The stages are as follows:

**Stage 1:**

$$
\begin{aligned}
BHAR = \quad & \beta_0 + \beta_1 \text{ firm age} + \beta_2 \text{ CSR score} + \beta_3 \text{ environmental malpractice disclosure} + \beta_4 \text{ leverage} \\
& + \beta_5 \text{ institutional ownership} + \beta_6 \text{ CRIO} + \beta_7 \text{ Z index} + \beta_8 \text{ ROA} + \beta_9 \text{ firm size} \\
& + \beta_{10} \text{ report attention} + \beta_{11} \text{ transparency} + \beta_{12} \text{ environmental subsidies} \\
& + \beta_{13} \text{ development capability} + \beta_{14} \text{ organizational slack} + \beta_{15} \text{ prior R\&D expenditure} \\
& + \beta_{16} \text{ prior political ties} + \beta_{17} \text{ TMT age heterogeneity} + \beta_{18} \text{ TMT education heterogeneity} \\
& + \beta_{19} \text{ TMT tenure heterogeneity} + \beta_{20} \text{ TMT occupation heterogeneity} + \beta_{21} \text{ CEO duality} \\
& + \beta_{22} \text{ stakeholder attention} + \beta_{23} \text{ Tobin's Q} + \beta_{24} \text{ local informal regulation} \\
& + \beta_{25} \text{ local pollution} + \beta_{26} \text{ local government fiscal power} + \beta_{27} \text{ sub industry} \\
& + \beta_{28} \text{ type of actual controller} + Residual
\end{aligned} \tag{5}
$$

**Stage 2:**

$$
\begin{aligned}
Residual = \quad & \beta_{00} + \beta_{01} \text{ polluting label} + \beta_{02} \text{ central ties} + \beta_{03} \text{ local ties} + \beta_{04} \text{ innovative capability} \\
& + \beta_{05} \text{ polluting label} * \text{ central ties} + \beta_{06} \text{ polluting label} * \text{ local ties} + \beta_{07} \text{ polluting label} \\
& * \text{ innovative capability} + e
\end{aligned} \tag{6}
$$

*Analysis of the interactive effects of political ties and innovative capability on investors' sense-making of the polluting label.* Another two-stage model was developed to examine the interactive effects of political ties and innovative capability on the investors' reactions, using the sample of 1670 NSM firms. The stages are as follows:

**Stage 1:**

$$
\begin{aligned}
BHAR = \quad & \beta_0' + \beta_1 \text{ firm age} + \beta_2 \text{ CSR score} + \beta_3 \text{ environmental malpractice disclosure} + \beta_4 \text{ leverage} \\
& + \beta_5 \text{ institutional ownership} + \beta_6 \text{ CRIO} + \beta_7 \text{ Z index} + \beta_8 \text{ ROA} + \beta_9 \text{ firm size} \\
& + \beta_{10} \text{ report attention} + \beta_{11} \text{ transparency} + \beta_{12} \text{ environmental subsidies} \\
& + \beta_{13} \text{ development capability} + \beta_{14} \text{ organizational slack} + \beta_{15} \text{ prior R\&D expenditure} \\
& + \beta_{16} \text{ prior political ties} + \beta_{17} \text{ TMT age Heterogeneity} + \beta_{18} \text{ TMT education heterogeneity} \\
& + \beta_{19} \text{ TMT tenure heterogeneity} + \beta_{20} \text{ TMT occupation heterogeneity} + \beta_{21} \text{ CEO duality} \\
& + \beta_{22} \text{ stakeholder attention} + \beta_{23} \text{ Tobin's Q} + \beta_{24} \text{ local informal regulation} \\
& + \beta_{25} \text{ local pollution} + \beta_{26} \text{ local government fiscal power} + \beta_{27} \text{ sub industry} \\
& + \beta_{28} \text{ type of actual controller} + Residual
\end{aligned} \tag{7}
$$

**Stage 2:**

$$
\begin{aligned}
Residual = \quad & \beta_{00} + \beta_{02} \text{ central ties} + \beta_{03} \text{ local ties} + \beta_{04} \text{ innovative capability} + \beta_{05} \text{ central ties} * \text{ local ties} \\
& + \beta_{05} \text{ central ties} * \text{ innovatapabilitylity} + \beta_{06} \text{ local ties} * \text{ innovatapabilitylity} + e
\end{aligned} \tag{8}
$$

## 4. Results

The prerequisite for using the DID (difference-in-differences) method is to have satisfied the parallel trends assumption. Following the approach of Beck et al. (2010) [52], this study utilizes both graphical and regression methods to examine the NSM policy parallel trends and dynamic effects.

Firstly, the graphical method is employed to compare the trends in enterprise value before and after receiving the polluting label. As depicted in Figure 1, there was no significant difference in urban innovation levels between the treatment and control groups before the polluting label was assigned, confirming the parallel trends. It is evident from Figure 1 that after enterprises received the polluting label, the policy effect emerged and strengthened over time. Furthermore, although 7 years after the introduction of the NSM the market value rebounded, the coefficients on $D^8$ to $D^{10}$ are not significant at the 5% level. Therefore, this study excludes the samples that were continuously monitored for more than 7 years, and retests them. Figure 2 indicates that the results are robust.

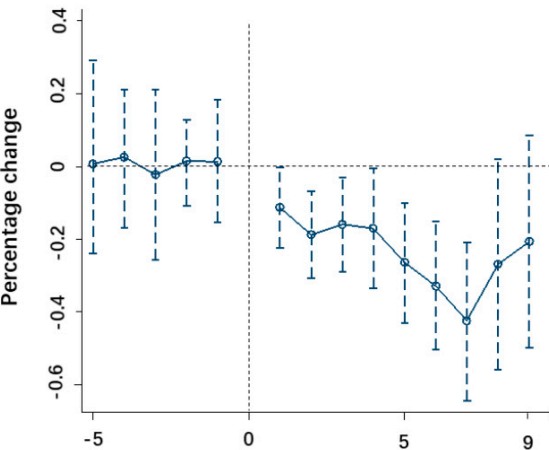

**Figure 1.** Dynamic impact of NSM program on market value. The figure plots the impact of NSM duration on the market value of NSM firms. A 15-year window, spanning from 5 years before NSM to 9 years on the list. The dashed lines represent 95% confidence intervals.

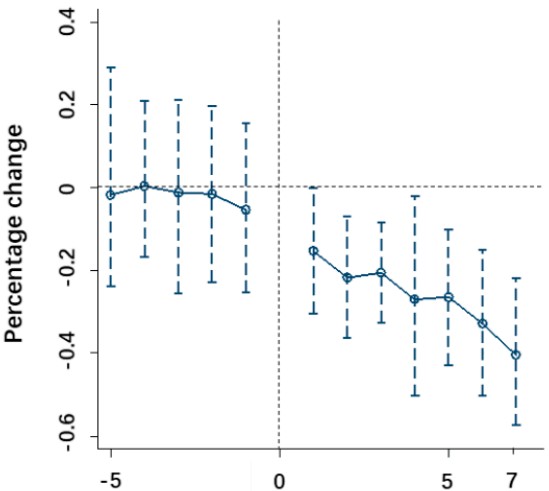

**Figure 2.** Dynamic impact of NSM program on market value. The figure plots the impact of NSM duration on the market value of NSM firms. A 12-year window, spanning from 5 years before NSM to 7 years on the list. The dashed lines represent 95% confidence intervals.

Subsequently, to further validate the parallel trends and assess the dynamic policy effects of the NSM, regression results are presented in Table 5. Table 5 indicates that, after being labeled as polluting, the policy coefficient was significantly different from 0, while before receiving the label, the policy coefficient was not significantly different from 0. This further demonstrates that the treatment and control city groups satisfied the parallel trends assumption before the policy implementation. Additionally, after the policy enactment, the absolute value of the regression coefficient increased annually. Consequently, the polluting label had a long-term and stable impact on enterprise value. Hypothesis 1 is thus validated.

This study then tests H2 and H3 using the two-stage model with a sample of 1670 NSM and 1670 non-NSM firms obtained by 1:1 year-by-year PSM. The coefficient of the polluting label ($\beta = -0.119$, $p < 0.01$) is negative and significant, verifying H1 again. The coefficients of the interactive terms are significant, implying that political ties and innovative capability exert different forces on NSM and non-NSM firms; the different effect coefficients are presented in Figure 3. In Figure 3, central ties ($\beta = 0.762$, $p < 0.01$) and innovative capability ($\beta = 0.202$, $p < 0.01$) trigger positive sense-making for investors, while the role of local ties ($\beta = 0.135$, $p < 0.01$) is the opposite. This conclusion is consistent with research conducted by Chaudhary et al. (2021) which showed a positive impact of central government associations

but that local ties tend to engender negative reactions [53]. These results demonstrate that H2a, H2b, and H3 are supported.

**Table 5.** Descriptive statistics and bivariate correlations.

| | **Coefficient** | | **Coefficient** |
|---|---|---|---|
| Before5 | 0.002 (0.166) | After3 | 0.179 (0.055) *** |
| Before4 | 0.010 (0.079) | After4 | 0.182 (0.072) ** |
| Before3 | −0.008 (0.159) | After5 | 0.264 (0.074) *** |
| Before2 | 0.006 (0.052) | After6 | 0.328 (0.081) *** |
| Before1 | 0.004 (0.070) | After7 | 0.415 (0.105) *** |
| Current | - | After8 | 0.266 (0.166) |
| After1 | 0.112 (0.053) ** | After9 | 0.201 (0.174) |
| After2 | 0.191 (0.052) *** | | |
| | Constant | | −0.936 (0.268) *** |
| | Obs. | | 3340 |
| | R2 | | 0.289 |

Notes. ** $p < 0.05$, *** $p < 0.001$.

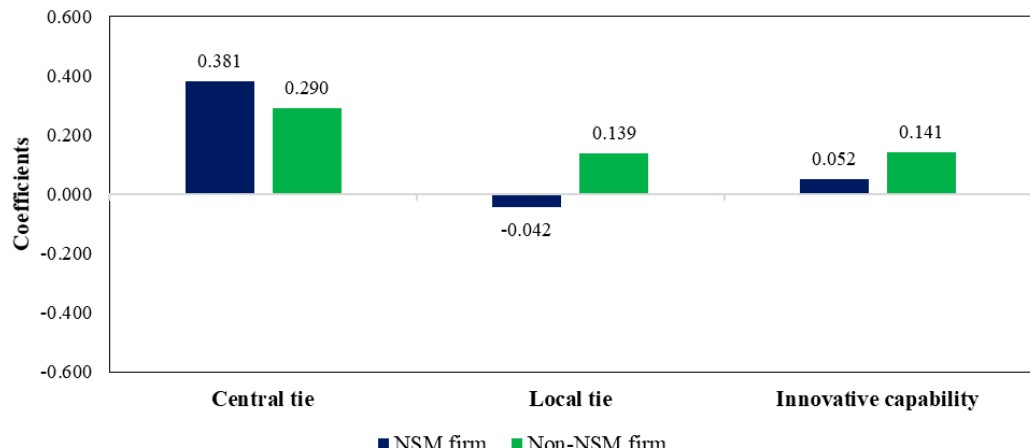

**Figure 3.** Comparison of NSM and non-NSM firms' coefficients of political ties and the impact of green innovative capability on investors' sense-making.

Table 6 shows the correlations of the variables, and Table 7 lists the results of the two-stage regression model. Stage 1 concerns the parameter estimates and t-statistics for the control variables alone, as shown in Figure 4. The results indicate that prior CSR performance, leverage, CRIO, firm size, report attention, stakeholder attention, and Tobin's Q induce positive investor sensemaking, aligning with research by Dhaliwal and Li (2011) which found that CSR initiatives play a pivotal role in swaying investor perceptions, especially during crisis situations [54]. In Stage 2, the effects of central ties, local ties, and innovative capability on the investors' reaction to firms with the polluting label are examined. Model 5 is our full model that includes all main and interaction effects. The coefficients of central ties (β = 0.461, $p < 0.01$) and innovative capability (β = 0.405, $p < 0.01$) are positive and significant, indicating that central ties and innovative capability trigger positive sense-making for investors. That is, H2a and H3 are again supported. The coefficient of local ties (β = −0.070, $p < 0.10$) is negative and significant, suggesting that local ties trigger investors' negative sense-making under the condition of the polluting label, and H2b is supported soundly. The coefficients for all interactions are significant.

**Table 6.** Descriptive statistics and bivariate correlations.

| NO. | Variable | Mean | Std.dev. | 1 | 2 | 3 | 4 | 5 | 6 | 7 | 8 | 9 | 10 | 11 |
|---|---|---|---|---|---|---|---|---|---|---|---|---|---|---|
| 1 | Market value | 0.738 | 5.997 | | | | | | | | | | | |
| 2 | Polluting label (dummy) | 0.500 | 0.500 | −0.055 ** | | | | | | | | | | |
| 3 | Central tie | 0.410 | 0.363 | 0.019 | 0.432 *** | | | | | | | | | |
| 4 | Local tie | 0.213 | 0.920 | 0.090 *** | −0.296 *** | −0.653 *** | | | | | | | | |
| 5 | Green innovative capability | 0.353 | 0.307 | −0.049 ** | 0.211 *** | 0.295 *** | −0.179 *** | | | | | | | |
| 6 | Firm age | 1.148 | 0.157 | 0.050 ** | 0.062 ** | −0.086 *** | 0.115 *** | −0.100 *** | | | | | | |
| 7 | CSR score | 2.47 | 1.664 | −0.03 | 0.113 *** | 0.064 ** | −0.03 | 0.073 ** | −0.056 ** | | | | | |
| 8 | Malpractice disclosure | 0.095 | 0.223 | −0.027 | 0.315 *** | 0.132 *** | −0.062 ** | 0.044 * | 0.187 *** | 0.012 | | | | |
| 9 | Leverage | 0.405 | 0.307 | 0.01 | 0.355 *** | 0.122 *** | −0.073 ** | 0.078 ** | 0.002 | −0.052 ** | 0.132 *** | | | |
| 10 | Institutional ownership | 0.208 | 0.219 | −0.008 | 0.070 ** | −0.078 ** | 0.050 ** | −0.006 | 0.069 ** | 0.039 | 0.112 *** | −0.01 | | |
| 11 | CRIO | 0.652 | 0.135 | −0.005 | −0.347 *** | −0.098 *** | 0.02 | −0.042 * | −0.239 *** | 0.159 *** | −0.165 *** | −0.152 *** | 0.144 *** | |
| 12 | Z index | 1.07 | 1.137 | −0.053 ** | 0.226 *** | 0.102 *** | −0.112 *** | 0.03 | 0.017 | −0.003 | 0.082 *** | 0.135 *** | −0.062 ** | −0.097 *** |
| 13 | ROA | 0.044 | 0.069 | 0 | −0.176 *** | −0.080 *** | 0.039 | 0.019 | −0.01 | 0.319 *** | −0.091 *** | −0.355 *** | 0.095 *** | 0.274 *** |
| 14 | Firm size | 9.421 | 0.466 | −0.096 *** | 0.471 *** | 0.191 *** | −0.117 *** | 0.142 *** | −0.067 ** | 0.231 *** | 0.324 *** | 0.442 *** | 0.137 *** | −0.044 * |
| 15 | Report Attention | 0.929 | 0.571 | −0.033 | −0.035 | 0.056 ** | −0.034 | 0.071 ** | −0.119 *** | 0.361 *** | 0.029 | −0.045 * | 0.124 *** | 0.226 *** |
| 16 | Transparency | 2.427 | 1.067 | 0.046 * | −0.203 *** | 0.012 | 0.014 | 0.012 | 0.018 | 0.055 ** | 0.033 | −0.208 *** | −0.027 | 0.108 *** |
| 17 | Environmental subsidies | 2.393 | 2.958 | −0.065 ** | 0.369 *** | 0.075 ** | −0.053 ** | 0.035 | 0.093 *** | −0.032 | 0.196 *** | 0.177 *** | 0.068 ** | −0.150 *** |
| 18 | Development capability | 0.167 | 0.886 | 0.004 | −0.007 | 0.014 | 0.004 | 0.017 | 0.021 | 0.012 | −0.026 | 0.043 * | 0.038 | 0.028 |
| 19 | Organizational slack | 6.744 | 4.433 | 0.015 | −0.144 *** | −0.073 ** | 0.041 * | 0.027 | −0.085 *** | 0.273 *** | −0.109 *** | −0.241 *** | 0.087 *** | 0.241 *** |
| 20 | Prior R&D expenditure | 0.42 | 0.042 | 0.093 *** | −0.389 *** | −0.181 *** | 0.145 *** | −0.125 *** | 0.029 | −0.061 ** | −0.111 *** | −0.266 *** | 0.029 | 0.111 *** |
| 21 | Prior Political tie | 0.13 | 0.095 | −0.015 | 0.168 *** | 0.171 *** | −0.03 | 0.580 *** | −0.028 | 0.074 ** | −0.029 | 0.042 * | −0.084 *** | −0.019 |
| 22 | TMT Age Heterogeneity | 0.178 | 0.045 | 0.056 ** | −0.412 *** | −0.083 *** | 0.087 *** | 0.014 | −0.026 | −0.090 *** | −0.175 *** | −0.209 *** | −0.085 *** | 0.135 *** |
| 23 | TMT Education Heterogeneity | 0.259 | 0.126 | 0.023 | −0.039 | −0.001 | −0.031 | 0.036 | −0.060 ** | 0.022 | 0.022 | −0.059 ** | 0.081 *** | 0.164 *** |
| 24 | TMT Tenure Heterogeneity | 0.388 | 0.284 | 0.036 | 0.192 *** | 0.067 ** | 0.041 * | −0.047 * | 0.291 *** | −0.084 *** | 0.230 *** | 0.176 *** | 0.064 ** | −0.335 *** |
| 25 | TMT Occupation Heterogeneity | 0.476 | 0.117 | 0.006 | 0.073 ** | 0.034 | 0.033 | 0.001 | 0.031 | −0.009 | 0.129 *** | 0.190 *** | −0.016 | −0.085 *** |
| 26 | CEO duality | 0.353 | 0.478 | 0.024 | −0.307 *** | −0.160 *** | 0.109 *** | −0.085 *** | −0.043 * | −0.080 *** | −0.114 *** | −0.140 *** | 0.017 | 0.099 *** |
| 27 | Stakeholder Attention | 4.475 | 4.981 | 0.02 | −0.298 *** | −0.160 *** | 0.028 | −0.081 *** | −0.133 *** | −0.062 ** | −0.136 *** | −0.131 *** | −0.044 * | 0.209 *** |
| 28 | Tobin's Q | 2.675 | 2.214 | 0.100 *** | −0.377 *** | −0.217 *** | 0.145 *** | −0.132 *** | 0.127 *** | −0.017 | −0.175 *** | −0.341 *** | 0.090 *** | 0.199 *** |
| 29 | Formal regulation | 4.289 | 0.307 | −0.03 | −0.109 *** | −0.107 *** | 0.095 *** | −0.045 * | 0.015 | −0.065 ** | −0.017 | −0.099 *** | −0.013 | 0.027 |
| 30 | Informal regulation | 3.193 | 0.435 | 0.001 | −0.089 *** | 0.007 | −0.011 | −0.011 | −0.147 *** | 0.052 ** | −0.097 *** | −0.082 *** | −0.158 *** | 0.091 *** |
| 31 | Local pollution | 1.684 | 0.273 | −0.003 | 0.094 *** | 0.060 ** | −0.03 | −0.003 | 0 | −0.019 | −0.016 | −0.059 ** | −0.129 *** | −0.062 ** |
| 32 | Local government fiscal power | 3.532 | 0.312 | 0.067 ** | −0.500 *** | −0.263 *** | 0.219 *** | −0.176 *** | 0.090 *** | −0.067 ** | −0.043 * | −0.124 *** | 0.072 ** | 0.204 *** |

| NO. | Variable | | | 12 | 13 | 14 | 15 | 16 | 17 | 18 | 19 | 20 | 21 | 22 |
|---|---|---|---|---|---|---|---|---|---|---|---|---|---|---|
| 13 | ROA | — | — | −0.063 ** | | | | | | | | | | |
| 14 | Firm size | — | — | 0.250 *** | −0.02 | | | | | | | | | |
| 15 | Report Attention | — | — | −0.033 | 0.382 *** | 0.348 *** | | | | | | | | |
| 16 | Transparency | — | — | −0.163 *** | 0.136 *** | −0.203 *** | 0.172 *** | | | | | | | |
| 17 | Environmental subsidies | — | — | 0.071 ** | −0.115 *** | 0.176 *** | −0.049 ** | −0.034 | | | | | | |
| 18 | Development capability | — | — | −0.036 | 0.097 *** | 0.003 | 0.060 ** | 0.011 | −0.041 * | | | | | |
| 19 | Organizational slack | — | — | −0.103 *** | 0.600 *** | 0.004 | 0.318 *** | 0.151 *** | −0.080 *** | 0.080 *** | | | | |
| 20 | Prior R&D expenditure | — | — | −0.134 *** | 0.091 *** | −0.179 *** | 0.065 ** | 0.197 *** | −0.196 *** | −0.023 | 0.101 *** | | | |
| 21 | Prior Political tie | — | — | −0.016 | 0.080 *** | 0.075 ** | 0.073 ** | 0.029 | 0.039 | 0.012 | 0.051 ** | −0.113 *** | | |
| 22 | TMT Age Heterogeneity | — | — | −0.148 *** | 0.078 ** | −0.334 *** | −0.062 ** | 0.121 *** | −0.134 *** | −0.002 | 0.080 *** | 0.101 *** | 0.050 ** | |
| 23 | TMT Education Heterogeneity | — | — | −0.014 | 0.019 | 0.027 | 0.063 ** | 0.219 *** | −0.017 | 0.075 ** | 0.058 ** | 0.114 *** | 0.009 | 0.012 |
| 24 | TMT Tenure Heterogeneity | — | — | −0.147 *** | −0.191 *** | 0.208 *** | −0.111 *** | 0.008 | 0.125 *** | −0.015 | −0.157 *** | 0.047 * | −0.060 ** | −0.147 *** |
| 25 | TMT Occupation Heterogeneity | — | — | 0.107 *** | −0.072 ** | 0.172 *** | 0.042 * | 0.028 | 0.065 ** | 0.017 | −0.084 *** | −0.009 | 0.043 * | −0.148 *** |
| 26 | CEO duality | — | — | −0.113 *** | 0.051 ** | −0.205 *** | −0.026 | 0.124 *** | −0.140 *** | 0.002 | 0.063 ** | 0.220 *** | −0.060 ** | 0.132 *** |
| 27 | Stakeholder Attention | — | — | −0.123 *** | 0.069 ** | −0.320 *** | −0.097 *** | −0.023 | −0.074 ** | −0.012 | 0.075 ** | −0.011 | −0.072 ** | 0.203 *** |
| 28 | Tobin's Q | — | — | −0.151 *** | 0.300 *** | −0.444 *** | −0.005 | 0.050 ** | −0.135 *** | 0.073 ** | 0.112 *** | 0.240 *** | −0.049 ** | 0.157 *** |
| 29 | Formal regulation | — | — | −0.111 *** | 0.042 * | −0.130 *** | 0.018 | 0.117 *** | 0.034 | −0.027 | 0.027 | 0.002 | −0.037 | 0.031 |
| 30 | Informal regulation | — | — | −0.080 *** | 0.050 ** | −0.163 *** | 0.093 *** | 0.165 *** | −0.033 | −0.003 | 0.064 ** | −0.038 | 0.022 | 0.042 * |
| 31 | Local pollution | — | — | −0.050 ** | −0.022 | −0.104 *** | −0.044 * | 0.125 *** | 0.085 *** | −0.021 | −0.036 | −0.062 ** | 0.003 | −0.035 |
| 32 | Local government fiscal power | — | — | −0.183 *** | 0.084 *** | −0.151 *** | 0.026 | 0.184 *** | −0.081 *** | 0.004 | 0.116 *** | 0.271 *** | −0.145 *** | 0.152 *** |

| NO. | Variable | | | 23 | 24 | 25 | 26 | 27 | 28 | 29 | 30 | 31 |
|---|---|---|---|---|---|---|---|---|---|---|---|---|
| 24 | TMT Tenure Heterogeneity | — | — | −0.110 *** | | | | | | | | |
| 25 | TMT Occupation Heterogeneity | — | — | 0.029 | 0.172 *** | | | | | | | |
| 26 | CEO duality | — | — | 0.041 * | −0.049 ** | −0.080 *** | | | | | | |
| 27 | Stakeholder Attention | — | — | 0.076 ** | −0.398 *** | −0.175 *** | 0.112 *** | | | | | |
| 28 | Tobin's Q | — | — | 0 | −0.120 *** | −0.067 ** | 0.123 *** | 0.267 *** | | | | |
| 29 | Formal regulation | — | — | 0.025 | −0.102 *** | −0.044 * | 0.074 ** | 0.087 ** | 0.018 | | | |
| 30 | Informal regulation | — | — | 0.058 ** | −0.323 *** | −0.051 ** | 0.072 ** | 0.219 *** | 0.076 ** | 0.551 *** | | |
| 31 | Local pollution | — | — | 0.035 | −0.070 ** | −0.091 *** | 0.031 | 0.016 | −0.088 *** | 0.754 *** | 0.570 *** | |
| 32 | Local government fiscal power | — | — | 0.101 *** | 0.084 *** | 0.022 | 0.229 *** | 0.068 ** | 0.170 *** | 0.537 *** | 0.240 *** | 0.267 *** |

Notes. * $p < 0.1$, ** $p < 0.05$, *** $p < 0.001$.

**Table 7.** Results of second stage regression.

| Variables | Model 1 | Model 2 | Model 3 | Model 4 | Model 5 |
|---|---|---|---|---|---|
| Intercept | 0.046 (0.016) ** | 0.043 (0.016) ** | 0.046 (0.016) ** | 0.047 (0.016) ** | 0.038 (0.016) ** |
| Central tie | 0.422 (0.060) *** | 0.439 (0.064) *** | 0.411 (0.060) *** | 0.424 (0.060) *** | 0.461 (0.081) *** |
| Local tie | −0.065 (0.003) *** | −0.057 (0.003) *** | −0.076 (0.003) *** | −0.062 (0.003) *** | −0.070 (0.004) *** |
| Green innovative capability | 0.421 (0.032) *** | 0.425 (0.034) *** | 0.442 (0.034) *** | 0.438 (0.032) *** | 0.405 (0.045) *** |
| Central tie × Local tie | | 0.307 (0.061) *** | | | 0.297(0.052) *** |
| Green innovative capability × Local tie | | | | 0.121 (0.077) * | 0.112 (0.067) * |
| Central tie × Green innovative capability | | | −0.284 (0.125) ** | | −0.326 (0.138) ** |
| Type of actual controller (dummy) | Included | Included | Included | Included | Included |
| Year (dummy) | Included | Included | Included | Included | Included |
| Model F-value | 16.52 *** | 15.38 *** | 13.20 *** | 14.20 *** | 10.07 *** |
| $R^2$ | 0.235 | 0.254 | 0.235 | 0.244 | 0.278 |
| Adjusted $R^2$ | 0.226 | 0.243 | 0.225 | 0.234 | 0.262 |

Notes. * $p < 0.1$, ** $p < 0.05$, *** $p < 0.001$.

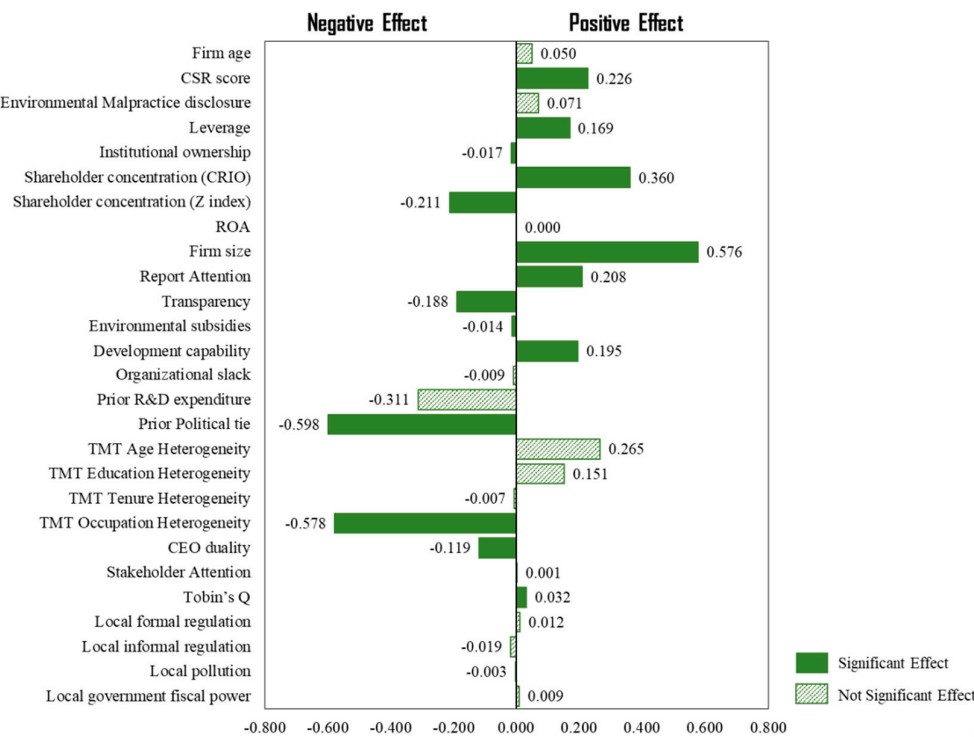

**Figure 4.** Coefficients of control variables in the first stage.

Figure 5 indicates a positive interaction between central and local ties on investors' sense-making of the polluting label (β = 0.297, $p < 0.05$). In support of H4a, investors have an increasingly positive sense-making when central ties are high, and negative sense-making (through strengthening of local ties) when they are low. That is, in the context of the polluting label, once the protection of central ties is lost, investors will be averse to loss. This is supportive of our argument that investors pay more attention to resources that bring certain and quick benefits.

Figure 6 indicates a positive interaction between local ties and innovative capability of investors' sense-making of the polluting label (β = 0.112, $p < 0.05$). When corporate innovative capability is high, the increase in the strength of local ties does not trigger further loss perception in investors. It demonstrates that for local ties, innovative capability serves as a buffer for investors' negative sense-making. Facing the loss prospects brought about by local ties, they are risk-hungry, hoping that high-return innovation capabilities can help them recover losses and even make profits, supporting H4a_1.

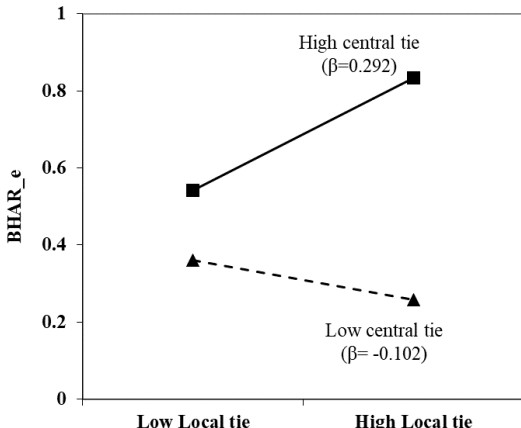

**Figure 5.** Impact of central and local ties on investors' sense-making.

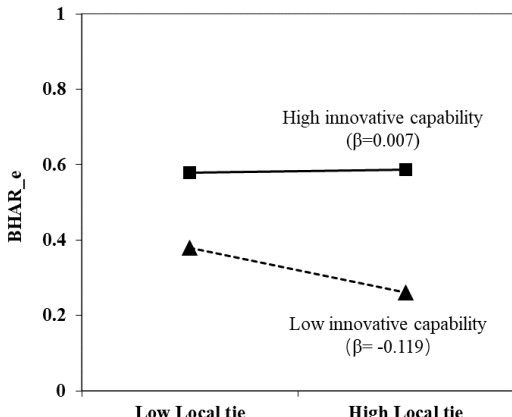

**Figure 6.** Impact of local ties and innovative capability on investors' sense-making.

Figure 7 indicates a negative interaction between central ties and innovative capability on investors' sense-making of the polluting label ($\beta = -0.326$, $p < 0.01$). As illustrated in Figure 7, the positive effect of innovative capability is weaker when central ties are strong and vice versa. These results confirm the inhibitory effect of central ties on innovation capability. That is, although both innovative capability and central ties positively regulate investors' negative perception of local ties, investors prefer central ties yielding fast returns and high certainty. Thus, H4c is supported.

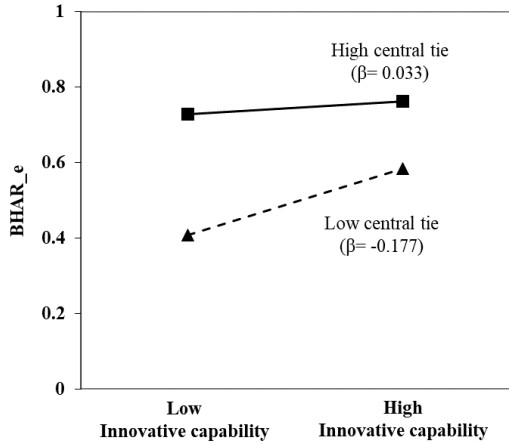

**Figure 7.** Impact of central ties and innovative capability on investors' sense-making.

## 5. Robustness Test

To ensure the reliability of our primary findings that suggest a negative association between the polluting label and investors' perception, this study executed several robustness checks. Initially employing a difference-in-differences (DID) approach, we assessed the impact of polluting labels on investor perception and discovered a significant decline in the value of companies listed on the NSM list. After applying a 1:1 propensity score matching (PSM), a comprehensive sample was derived. Subsequently, our treatment variable, which identified whether a company is on the NSM list or not, was tested in a two-stage model, revealing a notably negative impact. This outcome further substantiated the adverse effect of the polluting label on firms. To deepen our scrutiny, a mixed cross-sectional regression analysis of all NSM companies from 2016 to 2020 was conducted. The results are presented in Table 8. Consistent with our primary results, this method illustrated a positive correlation between central political ties and investors' perception (BHAR) with $\beta = 0.244$ ($p < 0.01$) and a similarly positive relationship with innovative capability with $\beta = 0.153$ ($p < 0.01$). However, local political ties displayed a negative association with investors' perception, as indicated by $\beta = -0.102$ ($p < 0.10$). Delving deeper, this study found positive effects on investors' perception of the polluting label from both the interaction between central and local ties ($\beta = 0.176$, $p < 0.05$) and the interaction between local ties and innovative capability ($\beta = 0.198$, $p < 0.05$). Yet, the interaction between central ties and innovative capability showcased a negative influence with $\beta = -0.260$ ($p < 0.01$). These robustness checks, taken collectively, corroborate the stability and consistency of our initial findings.

**Table 8.** Regression results of the robustness test.

| Variables | Model 1 | Model 2 | Model 3 | Model 4 | Model 5 |
|---|---|---|---|---|---|
| Intercept | 0.087 (0.022) *** | 0.088 (0.022) *** | 0.085 (0.022) *** | 0.087 (0.022) *** | 0.081 (0.023) *** |
| Central tie | 0.213 (0.065) *** | 0.221 (0.064) *** | 0.218 (0.065) *** | 0.214 (0.064) *** | 0.244 (0.074) *** |
| Local tie | −0.098 (0.051) * | −0.097 (0.051) * | −0.116 (0.052) * | −0.098 (0.051) * | −0.102 (0.052) * |
| Green innovative capability | 0.144 (0.035) *** | 0.147 (0.039) *** | 0.145 (0.033) *** | 0.171 (0.051) *** | 0.153 (0.033) *** |
| Central tie × Local tie | | 0.222 (0.088) ** | | | 0.176 (0.071) ** |
| Green innovative capability × Local tie | | | | 0.196 (0.074) ** | 0.198 (0.088) ** |
| Central tie × Green innovative capability | | | −0.299 (0.053) *** | | −0.260 (0.043) *** |
| Type of actual controller (dummy) | Included | Included | Included | Included | Included |
| Year (dummy) | Included | Included | Included | Included | Included |
| Model F-value | 18.70 *** | 18.88 *** | 18.12 *** | 12.14 *** | 12.44 *** |
| R² | 0.341 | 0.334 | 0.331 | 0.345 | 0.369 |
| Adjusted R² | 0.331 | 0.323 | 0.322 | 0.335 | 0.362 |

Notes. * $p < 0.1$, ** $p < 0.05$, *** $p < 0.001$.

In conclusion, our rigorous robustness tests consistently validate the initial findings. The polluting label bears a negative association with investors' perception, substantiating the sentiment that such labels pose a tangible detriment to firms. However, the strength and direction of this relationship are notably influenced by political ties and innovative capabilities. These insights not only underscore the importance of considering institutional relationships and internal firm competencies when interpreting investor behavior but also emphasize the multifaceted impact of external labels on firm valuation in the eyes of investors.

## 6. Discussion and Conclusions

In today's volatile business environment, organizations strive to showcase their unique standing to potential investors, emphasizing their unparalleled resource advantages [50,55,56]. Building on this premise, the exploration of this study delves into the nuanced way investors interpret the polluting label, shedding light on how intertwined corporate resources and capabilities influence these perceptions.

Firstly, our study accentuates the pivotal role of prospect theory in investor decision-making. As environments grow more intricate, investors, evolving into adept cue-seekers,

begin to integrate prospect theory. This allows them to meticulously analyze external uncertainties in tandem with internal corporate capabilities. Interestingly, this innovative application of prospect theory to dissect the polluting label unfurls a fresh paradigm in understanding investor behavior amidst uncertainties.

Furthermore, diverging from previous works that categorized political affiliations in a binary manner [20], this study provides a more textured understanding. By examining the strength and intricacies of these ties, our findings unravel the tug-of-war businesses often confront between central and local government affiliations. Such an approach augments our comprehension of the myriad ways these political linkages mold investor perceptions.

Transitioning to the realm of innovative capabilities, it is evident that, notwithstanding the challenges ushered in by the polluting label, robust corporate innovative prowess still resonates positively with investors. However, this favorable perception tends to diminish when juxtaposed with dominant central ties, echoing the timeless principle that inherent corporate vigor invariably garners investor appreciation.

On the practical front, firms grappling with the repercussions of a polluting label can glean valuable insights. Strengthening central political ties and bolstering innovative capabilities can serve as twin pillars to counterbalance and potentially recalibrate negative investor perceptions.

Yet, as with any comprehensive study, certain boundaries and constraints delimit our research. Our empirical focus on the polluting label primarily caters to scenarios with a stark demarcation between dominant and subordinate logics. Consequently, the extrapolation of our conclusions to other contexts mandates prudence and circumspection. Moreover, while our narrative predominantly revolves around political ties and innovative capabilities vis-à-vis the polluting label, the canvas remains expansive. Unraveling other potential organizational cues, like corporate reputation, might offer invaluable insights, especially when businesses find themselves at crossroads with conflicting institutional mandates.

In conclusion, our endeavor bridges the intricate interplay between investor perceptions, political dynamics, and corporate innovation within the polluting label context. As sustainability emerges as the clarion call of modern businesses, comprehending these intertwined dynamics assumes paramount importance.

**Author Contributions:** Conceptualization, L.L. and J.X.; Writing—original draft, L.L.; Writing—review & editing, M.W. and J.X. All authors have read and agreed to the published version of the manuscript.

**Funding:** This research was funded by the National Natural Science Foundation of China grant number 72004213, 71902178, 7234066.

**Institutional Review Board Statement:** Not applicable.

**Informed Consent Statement:** Not applicable.

**Data Availability Statement:** Not applicable.

**Conflicts of Interest:** The authors declare no conflict of interest.

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
