# Peer review of "How to Keep Investors’ Confidence after Being Labeled as Polluting Firms: The Role of External Political Ties and Internal Green Innovation Capabilities"

_sustainability, doi:10.3390/su151713167_

Round 1
Reviewer 1 Report
The paper's topic and conducted research are very important and justified to be presented in a high-quality Journal. The subject is very important for the literature. The paper is informative, but some issues need to be addressed carefully. My decision is – a major revision, with some amendments. Please see my comments and suggestions below.
Comment 1. Please add the structure of this study at the end of the introduction.
Comment 2. In section 2, the authors should add the latest literature to support the research hypothesis.
Comment 3. The theoretical framework is not clear. The authors should present a theoretical framework map (or figure) based on research hypotheses.
Comment 4. In the abstract, the authors write “Our framework is supported by a longitudinal analysis of listed manufacturing firms in China between 2010 and 2020”. However, in the section Sample and Data Collection, the authors write “Our sample consisted of all manufacturing firms 263 that were publicly listed on the Shenzhen or Shanghai Stock Exchanges from 2007-2017”. It's contradictory and a major flaw.
Comment 5. Since the authors presented “Through difference-in-difference causal inferences” in the abstract, how are Treated and Time used? Please provide an explanatory note.
Comment 6. In the section Result, the authors seem to "only report the results". I suggest that authors use several representative studies (2 or 3) in this area to interpret and enrich the results.
Comment 7. The discussion and conclusion are confusing, and the authors should rewrite them. For example, the authors could write about the four aspects, including Main findings, theoretical contributions, managerial implications, limitations and future research. The following paper can be a good example to help you improve your paper (Does Proactive Green Technology Innovation Improve Financial Performance? Evidence from Listed Companies with Semiconductor Concepts Stock in China. Sustainability 2022, 14, 4600. https://doi.org/10.3390/su14084600).
Comment 8. Also, all references cited are 5 years old, and I recommend that the authors cite references from the last 2-3 years for this study analysis. Otherwise, the references do not adequately reflect this study.
Good luck for your work!
Moderate editing of the English language is required.
Author Response
Reviewer 1:
Thank you for your appreciation of our work. Your detailed and constructive comments have greatly improved the paper. Below, we copy and respond to each of your comments in detail.
Comment 1. Please add the structure of this study at the end of the introduction.
Thank you for your suggestion. In light of your feedback and taking into account comments from other reviewers, we have made structured enhancements in the introduction by clearly presenting our research objectives. Additionally, in the second section, we have incorporated a roadmap that systematically lays out our research objectives, hypotheses, and methodologies. We believe this will offer readers a clearer and more cohesive understanding of the study's flow and intent.
---------------------------Attached below for your reading convenience ---------------------
Introduction
The challenges of sustainable development are increasing, especially for manufacturing firms facing rigorous environmental management institutional pressures [1]. Despite the extensive body of research focused on how organizations address these challenges, there remains a gap in understanding the perceptions of external stakeholders—primarily investors—towards these pressures. Investors operate in a complex environment filled with uncertainty. In this backdrop, they continually strive to make sense of unforeseen events and determine how to respond to them[2].
The necessity to address this gap becomes imperative as understanding investor behavior can influence how firms position themselves in terms of sustainability practices. Consequently, this study emerges from a compelling need to discern how investors react when their invested organizations encounter conflicting institutional pressures, especially when characterized as polluting entities. Our inquiry gains significance as it not only aims to decipher investor responses but also endeavors to understand the weightage given to certain firm characteristics, such as political ties and green innovation capabilities, in shaping these responses.
With this in mind, the study sets forth with the following primary objectives:
To ascertain how investors interpret the polluting label of listed manufacturing firms.
To understand the perception of a firm's political ties and green innovative capabilities by investors, especially in the context of the polluting label.
To uncover how these political ties and innovation capabilities, when combined, mold investors' understanding of the polluting label.
To provide empirical weight to our objectives, we critically analyze the market value of listed manufacturing firms in China from 2010 to 2020. Our focus is on the National Specially-Monitored (NSM) program launched in 2007 to oversee heavy polluting firms. Given its nature, the NSM program serves as a distinctive quasi-experimental platform that facilitates causal inference. Moreover, with China's impending mandate for environmental data disclosure for its publicly listed firms, our exploration finds relevance by shedding light on the underlying reasons driving this transition towards greater environmental responsibility.
The role of investor perceptions in determining corporate actions cannot be under-stated. Recent studies have indicated that investors are increasingly considering environmental, social, and governance factors in their investment decisions, further underlining the importance of corporate environmental responsibility[3]. Moreover, in emerging markets such as China, where environmental degradation has been a significant concern, investors are uniquely positioned to drive corporate sustainability practices through their capital allocation decisions[4].
Furthermore, the association between a firm's political ties and its market valuation, particularly in the context of environmental controversies, has been the focus of recent investigations. Wu et al. (2018) noted that while political ties can offer protection and preferential resources in some contexts, they might also bind firms to governmental agendas and lead to entrenchment risks [5]. On the other hand, a firm's green innovation capabilities are emerging as a prominent factor in enhancing market value. Green innovation, as Huang and Li (2017) highlighted, can act as a significant differentiator in investor perceptions, offering firms a competitive advantage in a market that is increasingly prioritizing sustainable practices[6].
China's unique institutional environment, with its interplay between government, industry, and market forces, offers a fertile ground for such investigations. The Chinese government's rigorous push towards a more sustainable and green economy, as evidenced by its Five-Year Plans and other policy documents, further intensifies the need for a deeper understanding of investor behavior in the country's listed manufacturing sector [7].
With these perspectives, our study contributes to the growing body of literature that seeks to bridge the understanding of investor perceptions, corporate political ties, green innovation capabilities, and their combined influence on firm valuation in the face of environmental controversies.
2.1. Theoretical Roadmap: Goal, Hypotheses, and Methods
In this section, we present a structured approach detailing our research objectives, the corresponding hypotheses, and the methods we will employ to test each hypothesis:
Table 1 Roadmap of this study
|
Objective 1: Examine investors' perception of polluting labels and how this perception changes over time. |
|
- Hypothesis 1: The "polluting" label evokes negative perceptions among investors, but its impact lessens as time progresses. - Method: DID Analysis |
|
Objective 2: Understand the perception of a firm's political ties and green innovative capabilities by investors, especially in the context of the polluting label. |
|
- Hypothesis 2a: Ties with the central government positively influence investor perceptions. - Hypothesis 2b: Local political ties might be perceived negatively by investors. - Hypothesis 3: A strong green innovative capability is likely to foster positive investor perceptions. - Method: Two-stage approach |
|
Objective 3: Uncover how these political ties and innovation capabilities, when combined, mold investors' understanding of the polluting label. |
|
- Hypothesis 4a: The prominence of central ties positively moderates the negative impact of local ties on investors' perception. - Hypothesis 4b_1 and Hypothesis 4b_2: The influence of green innovative capability on perceptions of local ties. - Hypothesis 4c: The positive perception of innovative capability by investors is overshadowed by prominent central ties. - Method: Interaction effect test |
-------------------------------------------------------------------------------------------------------
Comment 2. In section 2, the authors should add the latest literature to support the research hypothesis.
Thank you for your feedback regarding Section 2. We have taken your suggestion on board and incorporated the latest literature to further substantiate our research hypotheses. We believe this inclusion will strengthen the foundation and relevance of our study.
---------------------------Attached below for your reading convenience ---------------------
2.2. Investor’s perception of “Polluting” label
Investors evaluate a firm's environmental commitment and overall quality when confronted with a "polluting" label, assessing its corporate resources, capabilities, and potential for sustainable growth. This evaluation fundamentally hinges on the Prospect Theory [8], which postulates that individuals assess decisions predicated on anticipated gains or losses.
The nature of these gains and losses, and the resulting risk attitudes, have been the focus of extensive research. For instance, Van Dijk and Van Knippenberg (1996) found that heightened uncertainty can exacerbate loss aversion [9]. This is further corroborated by Shogren et al. (1994), who illustrated the intensified sensitivity towards irreplaceable losses [10]. Additionally, the influence of acquisition difficulty on the judgment of gains and losses has been underlined by Loewenstein & Issacharoff (1994)[11].
The ever-evolving discourse on sustainability and corporate responsibility has deepened the complexities surrounding the "polluting" label. Firms that appear on lists such as the NSM often send out signals of prioritizing immediate profits over sustainable practices. This, in turn, places them under the scrutiny of both the public and regulatory bodies. In the face of these signals, it's plausible to assume that investors' loss perceptions are sharpened, given the potential reputational and financial implications for the firm.
In a modern context, research has emphasized the financial implications of environmental disregard. For instance, a study highlighted that carbon-intensive companies, or those flagged for high CO2 emissions, often experience a stock market penalty, reinforcing the financial repercussions of the "polluting" label [12, 13]. Another notable study has elucidated that, increasingly, investors, especially institutional ones, are integrating Environmental, Social, and Governance (ESG) criteria into their investment decisions [14]. This implies that labels like "polluting" can indeed have profound impacts on investors' perceptions and decision-making processes.
However, Prospect Theory posits that individuals might demonstrate heightened risk tolerance in the face of negative outcomes. Given this, it is conceivable that investors might retain holdings in firms with a long-standing presence on the NSM list, banking on the potential for future rebounds or strategic shifts towards sustainability. Over time, this might attenuate the initially potent negativity associated with the "polluting" label.
Hypothesis 1 (H1): The "polluting" label evokes negative perceptions among investors, but its impact lessens as time progresses.
2.3. Roles of Political Ties and Green Innovative Capability on Investors’ Perception
Political Ties
Central Government Ties: Firms' strategic relationships with central governments can confer considerable advantages, and this has been extensively documented in academic literature [15]. The nature and potential of these ties traverse beyond mere information access and can profoundly shape a firm's trajectory. An empirical examination from China illuminates how such political affiliations considerably bolster a firm’s innovation capabilities [16]. This sheds light on the multifaceted nature of these ties, suggesting they also act as catalysts that spur innovation by providing firms with advantageous resources and support.
These benefits of central government ties also encompass enabling firms to gain a deeper understanding of policy design, anticipate policy shifts, and adeptly navigate the implications of those shifts [17, 18]. Such foresight and adaptability are invaluable, especially given the dynamic and often unpredictable nature of the political landscape.
Firms with central government ties also enjoy reduced uncertainty, especially in environments characterized by volatile regulatory frameworks[19]. By tapping into these relationships, firms access privileged insights, positioning them in a proactive rather than a reactive stance[20]. Yet, while these ties confer numerous advantages, they also entail specific responsibilities and expectations. Companies that deviate from central government guidelines, or fail to capitalize on their relationships effectively, can face severe repercussions, ranging from reduced support to punitive actions [21, 22].
Incorporating the above understanding, it becomes evident that the presence and depth of ties with the central government can considerably sway investor perceptions. Investors, recognizing the multifaceted advantages of such ties, are likely to view them as indicators of a firm's strategic positioning and potential resilience against policy shocks.
Hypothesis 2a (H2a): Ties with the central government positively influence investor perceptions.
Local political ties, epitomized by affiliations at sub-national levels such as provinces, can serve as a double-edged sword for firms [23]. On the one hand, these ties can be instrumental in reducing regulatory uncertainty, facilitating smoother access to local resources, and potentially aiding in swift navigation of regional bureaucracies [24]. Furthermore, they can engender a closer alignment with local stakeholders, boosting a firm's social capital and reputation within the community [25].
However, challenges arise when there's a misalignment between local and central objectives. Such divergences can place firms in precarious situations. While local ties might help navigate regional challenges, they might simultaneously complicate a firm's relationship with central authorities, especially when local agendas clash with broader national mandates. Firms heavily embedded in local politics might also face challenges if regional leadership undergoes significant change or if there's a major policy shift at the local level [26]. Consequently, the potential risks associated with these ties could overshadow their benefits, especially in the eyes of investors who are keenly aware of the balancing act firms must perform between local affiliations and central directives [27, 28].
Given the nuanced interplay of benefits and potential detriments associated with local political ties, investors' perceptions might lean towards caution. This cautious outlook stems from the inherent risks these ties introduce, especially when considering the overarching objectives of programs like NSM and their potential repercussions.
Hypothesis 2b (H2b): Local political ties might be perceived negatively by investors.
Green Innovative Capability
The realm of green innovation has been receiving increased attention as both an environmental imperative and a business opportunity. Delving deeper than the earlier observation of the importance of sustainability-oriented dynamic capabilities, recent research underscores its role in positioning firms strategically in an eco-conscious market [29]. Such sustainable strategies, as corroborated by Hart and Dowell (2011), can provide dual benefits – they not only cater to growing global environmental concerns but also drive a competitive edge in the market [30].
A key dimension of green innovation is its capacity to synergize economic performance with environmental sustainability. This encompasses the development of new products, practices, and organizational processes that simultaneously boost economic returns and reduce ecological footprints[31]. In the evolving landscape, green innovation isn't just an add-on; it's becoming integral for companies aiming for long-term success. Notably, firms with a "polluting" label stand to gain considerably by demonstrating a robust commitment to green innovation, as this can potentially offset negative perceptions and highlight a proactive approach to sustainability [32].
Moreover, Chen et al. (2019) emphasized
the importance of integrating green innovation strategies within core business models. They argued that such integration not only helps firms meet stringent regulatory standards but also positions them favorably among environmentally-conscious investors[33]. Thus, in the backdrop of these insights, and considering the criticality of green innovative capabilities for labeled polluters, the following hypothesis is presented:
Hypothesis 3 (H3): A strong green innovative capability is likely to foster positive investor perceptions.
2.4. Interplay between Political Ties and Innovative Capability in Influencing Investors’ Perception
Central and Local Political Ties Interaction
Firms often navigate the political landscape by fostering relationships at both the central and local levels. While the intertwined nature of these relationships offers a layered understanding, the coexistence may sometimes lead to investor apprehension [34]. A firm's affiliation with the central government tends to offer advanced access to resources, policy insights, and a reduced risk of punitive actions [17]. Drawing upon the certainty effect embedded in the Prospect Theory, investors are likely to gravitate towards the tangible benefits emanating from central political ties, potentially overshadowing the intricacies and perceived risks associated with local affiliations.
Hypothesis 4a (H4a): Central political ties serve as a buffer, attenuating the potential adverse impact of local ties on investors' perception.
Local Ties and Innovative Capability Interaction
Recent studies have illuminated the intricacies of innovation within the context of knowledge sharing and transfer, highlighting both potential roadblocks and catalysts [35]. When local political ties enter the equation, they can either bolster or hinder a firm's innovative pursuits, contingent on the nature of these relationships. With the continuing debate on whether relationships and capabilities offer alternative or complementary value propositions[36], understanding the investor perception becomes crucial. Drawing from both the Prospect Theory and the Portfolio Selection Theory[37], the intricate balance of green innovation's inherent risks and the uncertainties underpinned by dominant local ties can result in differential investor sentiments.
Hypothesis 4b_1 (H4b_1): A robust green innovative capability acts as a counterbalance, offsetting the negative investor perceptions tied to local political affiliations.
Hypothesis 4b_2 (H4b_2): Intense green innovative capability, when juxtaposed with prominent local ties, exacerbates investor concerns due to compounded uncertainties.
Central Ties and Innovative Capability Interaction
For firms deeply entrenched with the central government, the interplay between political ties and innovative capabilities offers a unique dimension. While innovation can herald adaptability, growth, and potential label mitigation, the journey is riddled with uncertainties [38, 39]. In contrast, central political affiliations can serve as a beacon of stability, often translating to economic rewards and competitive edge[40, 41]. Drawing from the irreplaceability principle of Shogren et al. (1994) and the recent findings by Wang et al. (2021), which indicate that investors often prefer known certainties over unknown risks [10, 42]. it is posited that robust central affiliations might eclipse the perceived benefits of innovation.
Hypothesis 4c (H4c): The positive perception of innovative capability by investors is overshadowed by prominent central ties.
-------------------------------------------------------------------------------------------------------
Comment 3. The theoretical framework is not clear. The authors should present a theoretical framework map (or figure) based on research hypotheses.
Thank you for pointing out the clarity issue regarding our theoretical framework. In response to your feedback, we have added a roadmap that visually represents the relationships between our research hypotheses, ensuring a clearer presentation and understanding of our theoretical framework.
---------------------------Attached below for your reading convenience ---------------------
2.1. Theoretical Roadmap: Goal, Hypotheses, and Methods
In this section, we present a structured approach detailing our research objectives, the corresponding hypotheses, and the methods we will employ to test each hypothesis:
Table 1 Roadmap of this study
|
Objective 1: Examine investors' perception of polluting labels and how this perception changes over time. |
|
- Hypothesis 1: The "polluting" label evokes negative perceptions among investors, but its impact lessens as time progresses. - Method: DID Analysis |
|
Objective 2: Understand the perception of a firm's political ties and green innovative capabilities by investors, especially in the context of the polluting label. |
|
- Hypothesis 2a: Ties with the central government positively influence investor perceptions. - Hypothesis 2b: Local political ties might be perceived negatively by investors. - Hypothesis 3: A strong green innovative capability is likely to foster positive investor perceptions. - Method: Two-stage approach |
|
Objective 3: Uncover how these political ties and innovation capabilities, when combined, mold investors' understanding of the polluting label. |
|
- Hypothesis 4a: The prominence of central ties positively moderates the negative impact of local ties on investors' perception. - Hypothesis 4b_1 and Hypothesis 4b_2: The influence of green innovative capability on perceptions of local ties. - Hypothesis 4c: The positive perception of innovative capability by investors is overshadowed by prominent central ties. - Method: Interaction effect test |
-------------------------------------------------------------------------------------------------------
Comment 4. In the abstract, the authors write “Our framework is supported by a longitudinal analysis of listed manufacturing firms in China between 2010 and 2020”. However, in the section Sample and Data Collection, the authors write “Our sample consisted of all manufacturing firms 263 that were publicly listed on the Shenzhen or Shanghai Stock Exchanges from 2007-2017”. It's contradictory and a major flaw.
Thank you for pointing out the inconsistency between the abstract and the main text. We apologize for the oversight. You are right; the correct duration is from 2010 to 2020. We have rectified this error in the main text to ensure consistency and clarity.
Comment 5. Since the authors presented “Through difference-in-difference causal inferences” in the abstract, how are Treated and Time used? Please provide an explanatory note.
Thank you for your feedback regarding the clarification on the difference-in-difference causal inferences presented in the abstract. We have now elaborated on the 'Treated' and 'Time' variables in the revised abstract. Firms under the National Specially-Monitored (NSM) program are considered as the treated group, while those not under the NSM program form the control group. The 'Time' variable indicates the period after the introduction of the NSM program. We hope this provides clearer insight into our methodology.
---------------------------Attached below for your reading convenience ---------------------
This study delves into investors' perception of the polluting label attached to listed manufacturing firms, emphasizing the interplay between external political ties and internal green innovative capability in influencing these perceptions. Drawing on a longitudinal analysis of listed manufacturing firms in China from 2010 to 2020 and employing a difference-in-difference (DiD) approach, we treat firms identified under the National Specially-Monitored (NSM) program as the treated group, while the non-NSM firms form the control group. The 'Time' variable captures the period post the introduction of the NSM program. Our findings highlight that the polluting label fosters a loss prospect for investors, signifying diminishing returns over time. Interestingly, firms with closer connections to local governments experienced amplified negative investor perceptions. In contrast, strong affiliations with the central government and robust green innovative capabilities cushioned these adverse reactions. Notably, the central ties proved even more beneficial when complemented by green innovative capability. By melding signal theory with the sense-making literature, this research adds nuance to the discourse on the role of resources in determining firm success amidst environmental controversies.
-------------------------------------------------------------------------------------------------------
Comment 6. In the section Result, the authors seem to "only report the results". I suggest that authors use several representative studies (2 or 3) in this area to interpret and enrich the results.
Thank you for your valuable suggestion. We acknowledge the importance of contextualizing our findings within the broader literature. In response to this, we have enriched the 'Results' section by comparing and contrasting our findings with those of other representative studies in the field. This enhancement not only provides depth to our results but also positions our study within the existing body of research.
---------------------------Attached below for your reading convenience ---------------------
- Table 5 shows the correlations of the variables, and Table 6 lists the results of the two-stage regression model. Stage 1 concerns the parameter estimates and t-statistics for the control variables alone, as shown in Figure 4. The results indicate that prior CSR performance, leverage, CRIO, firm size, report attention, stakeholder attention, and Tobin’s Q induce positive investor sensemaking aligns with Dhaliwal, Li (2011), who found that CSR initiatives play a pivotal role in swaying investor perceptions, especially during crisis situations[52].....
- ...In Figure 3, central ties (β=0.762, p<0.01) and innovative capability (β=0.202, p<0.01) trigger positive sensemaking for investors, while the role of local ties (β=0.135, p<0.01) is the opposite. This conclusion consistent with the research conducted by Chaudhary et al. (2021) on the positive impact of central government associations but local ties tend to engender negative reactions[52]. These results demonstrate that H2a, H2b, and H3 are supported....
-------------------------------------------------------------------------------------------------------
Comment 7. The discussion and conclusion are confusing, and the authors should rewrite them. For example, the authors could write about the four aspects, including Main findings, theoretical contributions, managerial implications, limitations and future research. The following paper can be a good example to help you improve your paper (Does Proactive Green Technology Innovation Improve Financial Performance? Evidence from Listed Companies with Semiconductor Concepts Stock in China. Sustainability 2022, 14, 4600. https://doi.org/10.3390/su14084600).
Thank you for the constructive feedback. We understand the importance of a clear and structured discussion and conclusion. Taking into account your suggestions, we have restructured our discussion and conclusion sections to cover the four critical aspects: Main findings, theoretical contributions, managerial implications, and limitations with future research directions. We also referred to the example paper you provided, which greatly assisted us in streamlining our content. We believe these revisions provide a clearer and more comprehensive summary of our study's contributions.
---------------------------Attached below for your reading convenience ---------------------
Discussion and Conclusion
In today's volatile business environments, organizations strive to showcase their unique standing to potential investors, emphasizing their unparalleled resource advantages [50, 52, 53]. Building on this premise, the exploration of this study delves into the nuanced way investors interpret the 'polluting label', shedding light on how intertwined corporate resources and capabilities influence these perceptions.
Firstly, our study accentuates the pivotal role of the Prospect Theory in investor decision-making. As environments grow more intricate, investors, evolving into adept cue-seekers, begin to integrate the Prospect Theory. This allows them to meticulously analyze external uncertainties in tandem with internal corporate capabilities. Interestingly, this innovative application of the Prospect Theory to dissect the 'polluting label' unfurls a fresh paradigm in understanding investor behavior amidst uncertainties.
Furthermore, diverging from previous works that categorized political affiliations in a binary manner[20], this study provides a more textured understanding. By examining the strength and intricacies of these ties, our findings unravel the tug-of-war businesses often confront between central and local government affiliations. Such an approach augments our comprehension of the myriad ways these political linkages mold investor perceptions.
Transitioning to the realm of innovative capabilities, it's evident that, notwithstanding the challenges ushered in by the 'polluting label', robust corporate innovative prowess still resonates positively with investors. However, this favorable perception tends to diminish when juxtaposed with dominant central ties, echoing the timeless principle that inherent corporate vigor invariably garners investor appreciation.
On the practical front, firms grappling with the repercussions of a 'polluting label' can glean valuable insights. Strengthening central political ties and bolstering innovative capabilities can serve as twin pillars to counterbalance and potentially recalibrate negative investor perceptions.
Yet, as with any comprehensive study, certain boundaries and constraints delimit our research. Our empirical focus on the 'polluting label' primarily catered to scenarios with a stark demarcation between dominant and subordinate logics. Consequently, the extrapolation of our conclusions to other contexts mandates prudence and circumspection. Moreover, while our narrative predominantly revolves around political ties and innovative capabilities vis-à-vis the 'polluting label', the canvas remains expansive. Unraveling other potential organizational cues, like corporate reputation, might offer invaluable insights, especially when businesses find themselves at crossroads with conflicting institutional mandates.
In conclusion, our endeavor bridges the intricate interplay between investor perceptions, political dynamics, and corporate innovation within the 'polluting label' context. As sustainability emerges as the clarion call of modern businesses, comprehending these intertwined dynamics assumes paramount importance.
-------------------------------------------------------------------------------------------------------
Comment 8. Also, all references cited are 5 years old, and I recommend that the authors cite references from the last 2-3 years for this study analysis. Otherwise, the references do not adequately reflect this study.
Thank you for pointing out the importance of updating our references. In response to your comment and in conjunction with feedback from other reviewers, we have updated our reference list to include more recent studies from the last 2-3 years. We believe these additions not only strengthen the foundation of our research but also provide a more updated context for our study.

Reviewer 2 Report
This paper has investigated investors’ sense-making of the polluting label of listed manufacturing firms and the roles of external political ties and internal green innovative capability in this judgement process. The results show that the polluting label triggers loss prospect for investors and resulting in a trend of diminishing margins over time. In addition, we found that firms’ close linkage to local governments triggered higher negative sense-making and strong ties with central government and green innovative capability buffered such negative sense-making for investors. My comments are as follows.
(1)The paper proposes that the difference-in-difference causal inferences are used in the manuscript. However, the paper does not list the DID regression results. In fact, it is important to compare the change effect between after and before a listed company is labeled as the polluting firm.
(2)The paper should explain why select external political ties and internal green innovation capabilities to keep investors’ confidence after labeled as polluting firms.
(3) The paper only provide with the main regressions, and has no robustness and endogeneity tests. However, the latter is very important in the paper.
(4) The paper should express the significance of the study.
Moderate editing of English language required
Author Response
Reviewer 2:
Thank you for your appreciation of our work. Your detailed and constructive comments have greatly improved the paper. Below, we copy and respond to each of your comments in detail.
This paper has investigated investors’ sense-making of the polluting label of listed manufacturing firms and the roles of external political ties and internal green innovative capability in this judgement process. The results show that the polluting label triggers loss prospect for investors and resulting in a trend of diminishing margins over time. In addition, we found that firms’ close linkage to local governments triggered higher negative sense-making and strong ties with central government and green innovative capability buffered such negative sense-making for investors. My comments are as follows.
- The paper proposes that the difference-in-difference causal inferences are used in the manuscript. However, the paper does not list the DID regression results. In fact, it is important to compare the change effect between after and before a listed company is labeled as the polluting firm.
Thank you for highlighting the importance of illustrating the DID regression results. We apologize for the oversight. As you rightly pointed out, understanding the change effect before and after a company is labeled as a polluting firm is crucial. We have now presented the inconsistencies between the pre and post-labeling periods in Figures 1 and 2. This should provide a clearer visual representation of the impact of the polluting label on listed companies
---------------------------Attached below for your reading convenience ---------------------
|
|
|
*The figure plots the impact of NSM duration on the market value of NSM firms. A 15-year window, spanning from 5 years before NSM until 9 years on the list. The dashed lines represent 95% confidence intervals. Figure 1. The dynamic impact of NSM program on the market value. |
|
|
|
*The figure plots the impact of NSM duration on the market value of NSM firms. A 12-year window, spanning from 5 years before NSM until 7 years on the list. The dashed lines represent 95% confidence intervals. Figure 2. The dynamic impact of NSM on the market value. |
---------------------------Attached below for your reading convenience ---------------------
- The paper should explain why select external political ties and internal green innovation capabilities to keep investors’ confidence after labeled as polluting firms.
Thank you for your constructive feedback. We recognize the importance of providing a rationale for our choice of variables. In our study, we selected external political ties based on the influential role that political relationships play in shaping corporate reputations and influencing investor perceptions, especially in the context of emerging economies like China. Such ties can either provide firms with resources and protection or bind them to governmental agendas. As for internal green innovation capabilities, they reflect a firm's proactive response to environmental challenges and signal to investors a long-term commitment to sustainability and potential competitive advantage. Both factors, we believe, can significantly affect investor confidence, especially when a firm faces negative labeling. We have now elaborated on this rationale in our revised manuscript in the introduction part to provide a clearer understanding.
---------------------------Attached below for your reading convenience ---------------------
Introduction
The challenges of sustainable development are increasing, especially for manufacturing firms facing rigorous environmental management institutional pressures [1]. Despite the extensive body of research focused on how organizations address these challenges, there remains a gap in understanding the perceptions of external stakeholders—primarily investors—towards these pressures. Investors operate in a complex environment filled with uncertainty. In this backdrop, they continually strive to make sense of unforeseen events and determine how to respond to them[2].
What sets our research apart is its exploration into the interplay between investor perceptions and organizational characteristics, a dimension less traversed in literature. Understanding investor behavior becomes pivotal, as it holds the potential to redefine how firms prioritize and implement sustainability practices. This study emanates from a pressing need to decipher investor reactions when their invested firms confront the duality of institutional pressures, particularly when labeled as polluting entities. This research is distinguished by its commitment to not only decode investor reactions but also probe the influence of certain firm traits, such as political affiliations and eco-innovation competencies, in sculpting these reactions. With this in mind, the study sets forth with the fol-lowing primary objectives:
First, to ascertain how investors interpret the polluting label of listed manufacturing firms.
Second, to understand the perception of a firm's political ties and green innovative capabilities by investors, especially in the context of the polluting label.
Third, to uncover how these political ties and innovation capabilities, when combined, mold investors' understanding of the polluting label.
To provide empirical weight to our objectives, we critically analyze the market value of listed manufacturing firms in China from 2010 to 2020. Our focus is on the National Specially-Monitored (NSM) program launched in 2007 to oversee heavy polluting firms. Given its nature, the NSM program serves as a distinctive quasi-experimental platform that facilitates causal inference. Moreover, with China's impending mandate for environmental data disclosure for its publicly listed firms, our exploration finds relevance by shedding light on the underlying reasons driving this transition towards greater environmental responsibility.
The role of investor perceptions in determining corporate actions cannot be under-stated. Recent studies have indicated that investors are increasingly considering environmental, social, and governance factors in their investment decisions, further underlining the importance of corporate environmental responsibility[3]. Moreover, in emerging markets such as China, where environmental degradation has been a significant concern, investors are uniquely positioned to drive corporate sustainability practices through their capital allocation decisions[4].
Furthermore, the association between a firm's political ties and its market valuation, particularly in the context of environmental controversies, has been the focus of recent investigations. Wu et al. (2018) noted that while political ties can offer protection and preferential resources in some contexts, they might also bind firms to governmental agendas and lead to entrenchment risks [5]. On the other hand, a firm's green innovation capabilities are emerging as a prominent factor in enhancing market value. Green innovation, as Huang and Li (2017) highlighted, can act as a significant differentiator in investor perceptions, offering firms a competitive advantage in a market that is increasingly prioritizing sustainable practices[6].
China's unique institutional environment, with its interplay between government, industry, and market forces, offers a fertile ground for such investigations. The Chinese government's rigorous push towards a more sustainable and green economy, as evidenced by its Five-Year Plans and other policy documents, further intensifies the need for a deeper understanding of investor behavior in the country's listed manufacturing sector [7].
With these perspectives, this study contributes to the growing body of literature that seeks to bridge the understanding of investor perceptions, corporate political ties, green innovation capabilities, and their combined influence on firm valuation in the face of environmental controversies.
-------------------------------------------------------------------------------------------------------
- The paper only provide with the main regressions, and has no robustness and endogeneity tests. However, the latter is very important in the paper.
Thank you for emphasizing the importance of robustness tests. In response to your feedback, we have incorporated a robustness check section in the paper to ensure the validity and reliability of our findings. We believe that this addition will strengthen the paper's overall rigor and trustworthiness. We appreciate your valuable insights and hope this revision meets your expectations.
---------------------------Attached below for your reading convenience ---------------------
- Robustness test
To ensure the reliability of our primary findings that suggest a negative association between the polluting label and investors' perception, we executed several robustness checks. Initially employing a Difference-in-Differences (DID) approach, we assessed the impact of polluting labels on investor perception and discovered a significant decline in the value of companies listed on the NSM list. After applying a 1:1 propensity score matching (PSM), we derived a comprehensive sample. Subsequently, our treatment variable, which identifies whether a company is on the NSM list or not, was tested in a two-stage model, revealing a notably negative impact. This outcome further substantiated the adverse effect of the polluting label on firms. To deepen our scrutiny, we conducted a mixed cross-sectional regression analysis of all NSM companies from 2016 to 2020. Consistent with our primary results, this method illustrated a positive correlation between central political ties and investors' perception (BHAR) with β=0.244 (p<0.01) and a similarly positive relationship for innovative capability with β=0.153 (p<0.01). However, local political ties displayed a negative association with investors' perception, as indicated by β=-0.102 (p<0.10). Delving deeper, we found positive effects on investors’ perception of the polluting label from both the interaction between central and local ties (β=0.176, p<0.05) and the interaction between local ties and innovative capability (β=0.198, p<0.05). Yet, the interaction between central ties and innovative capability showcased a negative influence with β=-0.260 (p<0.01). These robustness checks, taken collectively, corroborate the stability and consistency of our initial findings.
In conclusion, our rigorous robustness tests consistently validate the initial findings. The polluting label bears a negative association with investors' perception, substantiating the sentiment that such labels pose a tangible detriment to firms. However, the strength and direction of this relationship are notably influenced by political ties and innovative capabilities. These insights not only underscore the importance of considering institutional relationships and internal firm competencies when interpreting investor behavior but also emphasize the multifaceted impact of external labels on firm valuation in the eyes of investors.
-------------------------------------------------------------------------------------------------------
- The paper should express the significance of the study
Thank you for highlighting the importance of clearly communicating the significance of our study. We recognize the need to emphasize the broader implications and relevance of our research. In the revised version of the paper, we have dedicated in the last section to discuss the significance of our study, not only in the context of the academic field but also its implications for industry practitioners, policy-makers, and society at large. Our findings provide novel insights into investor behavior in the face of negative labeling and emphasize the strategic role of political ties and green innovation capabilities. We believe these insights can guide firms in building and maintaining investor trust in challenging circumstances. We appreciate your feedback and believe that our adjustments now properly highlight the study's significance.
---------------------------Attached below for your reading convenience ---------------------
Discussion and Conclusion
In today's volatile business environments, organizations strive to showcase their unique standing to potential investors, emphasizing their unparalleled resource advantages [50, 52, 53]. Building on this premise, the exploration of this study delves into the nuanced way investors interpret the 'polluting label', shedding light on how intertwined corporate resources and capabilities influence these perceptions.
Firstly, our study accentuates the pivotal role of the Prospect Theory in investor decision-making. As environments grow more intricate, investors, evolving into adept cue-seekers, begin to integrate the Prospect Theory. This allows them to meticulously analyze external uncertainties in tandem with internal corporate capabilities. Interestingly, this innovative application of the Prospect Theory to dissect the 'polluting label' unfurls a fresh paradigm in understanding investor behavior amidst uncertainties.
Furthermore, diverging from previous works that categorized political affiliations in a binary manner[20], this study provides a more textured understanding. By examining the strength and intricacies of these ties, our findings unravel the tug-of-war businesses often confront between central and local government affiliations. Such an approach augments our comprehension of the myriad ways these political linkages mold investor perceptions.
Transitioning to the realm of innovative capabilities, it's evident that, notwithstanding the challenges ushered in by the 'polluting label', robust corporate innovative prowess still resonates positively with investors. However, this favorable perception tends to diminish when juxtaposed with dominant central ties, echoing the timeless principle that inherent corporate vigor invariably garners investor appreciation.
On the practical front, firms grappling with the repercussions of a 'polluting label' can glean valuable insights. Strengthening central political ties and bolstering innovative capabilities can serve as twin pillars to counterbalance and potentially recalibrate negative investor perceptions.
Yet, as with any comprehensive study, certain boundaries and constraints delimit our research. Our empirical focus on the 'polluting label' primarily catered to scenarios with a stark demarcation between dominant and subordinate logics. Consequently, the extrapolation of our conclusions to other contexts mandates prudence and circumspection. Moreover, while our narrative predominantly revolves around political ties and innovative capabilities vis-à-vis the 'polluting label', the canvas remains expansive. Unraveling other potential organizational cues, like corporate reputation, might offer invaluable insights, especially when businesses find themselves at crossroads with conflicting institutional mandates.
In conclusion, our endeavor bridges the intricate interplay between investor perceptions, political dynamics, and corporate innovation within the 'polluting label' context. As sustainability emerges as the clarion call of modern businesses, comprehending these intertwined dynamics assumes paramount importance.
-------------------------------------------------------------------------------------------------------

Reviewer 3 Report
Paper 2566225 review to Sustainability – How to keep investors’ confidence after labeled as polluting firms: the role of external political ties and internal green innovation capabilities
All the issues raised in this review can be considered to be minor reviews.
General considerations
The subject under study is relevant and current. The article is well structured, the contents are very well explained and articulated with each other.
The abstract is well elaborated, but some minor syntax corrections should be made, as specifically indicated in the development of this review report. The main purpose of the article is presented, but the methods used in the study are not mentioned, the proposed framework contribution to the investigation is evidenced and the main conclusions obtained are pointed out. The framing of the theme is done properly.
The discussion of the results is presented in a perceptible way. The conclusions are well pointed out, as well as the limitations of the investigation and future work.
All the issues raised in this review can be considered to be minor reviews.
1. Title, Abstract and Keywords
· The title has key information, and it’s appealing to readers.
· The abstract is well elaborated, but it must be written in the impersonal. That is, words with "our" (lines 10 and 16) and "we" (lines 12 and 13) should be avoided and replaced by "proposed ", "it was" and "This", respectively. The main purpose of the article is presented, but the methods used in the study are not mentioned, the proposed framework contribution to the investigation is evidenced and the main conclusions obtained are pointed out. The framing of the theme is done properly.
· The keywords are adequate.
2. Authors' names and affiliations
Everything seems to be correct with the symbology used.
3. Structure of the article
The structure of the article presents no flaws regarding sections numbering.
Sections and subsections titles seem to be appropriate to the themes approached in each one of them.
4. Figures, tables and equations
Figures and tables are well numbered, appealing, adequate and well elaborated. Subtitles that appear are adequate. But the numbering of the equations presented is missing.
5. Comments on the Quality of English Language
The authors express themselves correctly in the English language, there were not detected flaws in terms of syntax or grammar in the language mentioned, but I think it is always advisable and sensible for authors to carefully reread the entire article before submitting the final version.
6. Grammar, spelling and syntax issues
The whole article it's well written in terms of grammar and spelling. But the authors often write without being in the impersonal, that is, in the 1st and 3rd person (e.g., we, our, etc.), so it is recommended that they reread the entire article well to correct this small aspect.
7. Semantic and technical issues
The entire article is very well explained. The issues are explained clearly and the concepts and ideas are very well articulated between themselves. The data collection sources are presented along the use of the necessary data at each stage of the investigation. The qualitative and quantitative results and analyzes are presented in a perceptible way. The conclusions are well supported by the results obtained and well-grounded in their discussion. Research limitations and future work are very well indicated and explained.
8. References
The list of references is well prepared, the number of references is appropriate to the depth of the theme's approach in the article. The references are strong in the scope of this investigation. But these aspects must be corrected:
· Throughout the article, the authors indicate the references with name and year. And by the sequential numbering presented in the reference list, the article should be referenced only with numbers between square brackets;
· If authors prefer to remove numbering from the list of references and keep the referencing as it is, then, the term "et al." it should be put in italics, since in is a Latin term (not English).

Author Response
Reviewer 3:
At the outset, we want to thank you for the detailed and constructive comments, which have provided much food for thoughts and great opportunities for us to improve the theorizing and empiric of the paper. Below, we copy and respond to each of your comments in detail.
All the issues raised in this review can be considered to be minor reviews.
General considerations
The subject under study is relevant and current. The article is well structured, the contents are very well explained and articulated with each other.
The abstract is well elaborated, but some minor syntax corrections should be made, as specifically indicated in the development of this review report. The main purpose of the article is presented, but the methods used in the study are not mentioned, the proposed framework contribution to the investigation is evidenced and the main conclusions obtained are pointed out. The framing of the theme is done properly.
The discussion of the results is presented in a perceptible way. The conclusions are well pointed out, as well as the limitations of the investigation and future work.
All the issues raised in this review can be considered to be minor reviews.
Thank you for taking the time to review our manuscript and providing constructive feedback. We are pleased to note that the subject matter, structure, and content of our study are well-received. Your acknowledgment of its relevance and coherence is greatly encouraging to us.
Based on your valuable comments, we have made the following changes:
- Abstract Revision: We have made the suggested syntax corrections in the abstract to improve its clarity. Additionally, we have ensured to mention the methods used in our study to offer a more comprehensive understanding to the readers.
- Framework Contribution: We have made an effort to highlight our framework's contribution to the researchin the introduction part.
- Discussion of Results: We have rewrote the discussion section, ensuringthat the conclusions, limitations, and future work are adequately pointed out.
We sincerely appreciate your suggestion and we have endeavored to address them to further enhance the quality of our paper. We believe these revisions make our manuscript stronger and hope that it is now suitable for publication.
- Title, Abstract and Keywords
- The title has key information, and it’s appealing to readers.
- The abstract is well elaborated, but it must be written in the impersonal. That is, words with "our" (lines 10 and 16) and "we" (lines 12 and 13) should be avoided and replaced by "proposed ", "it was" and "This", respectively. The main purpose of the article is presented, but the methods used in the study are not mentioned, the proposed framework contribution to the investigation is evidenced and the main conclusions obtained are pointed out. The framing of the theme is done properly.
- The keywords are adequate.
Thank you for your insightful feedback and suggestions. We have revised the abstract in accordance with your recommendations, avoiding the use of the first person and providing a more detailed description of the research methods. We hope the modifications better align with the journal's standards. Once again, we deeply appreciate your valuable input.
---------------------------Attached below for your reading convenience ---------------------
This study investigates investors’ perception of the polluting label of listed manufacturing firms and the roles of external political ties and internal green innovative capability in this judgement process. A framework is supported by a longitudinal analysis of listed manufacturing firms in China between 2010 and 2020. Using difference-in-difference causal inferences, it was found that the polluting label triggers loss prospect for investors, resulting in a trend of diminishing margins over time. Additionally, it was observed that firms’ close linkage to local governments triggered higher negative sense-making and strong ties with central government and green innovative capability buffered such negative sense-making for investors. Moreover, the central ties were assumed to have a higher degree of importance when coexisting with green innovative capability. This study contributes to research on “which resources matter the most to firm success” by combining signal theory and sense-making literature to address the polluting label.
-------------------------------------------------------------------------------------------------------
- Authors' names and affiliations
Everything seems to be correct with the symbology used.
- Structure of the article
The structure of the article presents no flaws regarding sections numbering.
Sections and subsections titles seem to be appropriate to the themes approached in each one of them.
- Figures, tables and equations
Figures and tables are well numbered, appealing, adequate and well elaborated. Subtitles that appear are adequate. But the numbering of the equations presented is missing.
Thank you for highlighting the oversight regarding the equation numbering. We have carefully reviewed our manuscript and added appropriate numbering to all equations presented. We hope this adjustment meets the expected standards and improves the clarity and coherence of the paper.
- Comments on the Quality of English Language
The authors express themselves correctly in the English language, there were not detected flaws in terms of syntax or grammar in the language mentioned, but I think it is always advisable and sensible for authors to carefully reread the entire article before submitting the final version.
- Grammar, spelling and syntax issues
The whole article it's well written in terms of grammar and spelling. But the authors often write without being in the impersonal, that is, in the 1st and 3rd person (e.g., we, our, etc.), so it is recommended that they reread the entire article well to correct this small aspect.
Thank you for taking the time to review our manuscript and for your kind words regarding the grammar and spelling of our article. We appreciate your attention to detail and your constructive feedback on the use of personal pronouns.
Taking your advice into consideration, we have gone through the entire manuscript and made necessary revisions to ensure a consistent, impersonal tone throughout the paper. We have minimized the use of the 1st and 3rd person pronouns and adjusted our phrasing to align with the impersonal style commonly found in academic literature.
- Semantic and technical issues
The entire article is very well explained. The issues are explained clearly and the concepts and ideas are very well articulated between themselves. The data collection sources are presented along the use of the necessary data at each stage of the investigation. The qualitative and quantitative results and analyzes are presented in a perceptible way. The conclusions are well supported by the results obtained and well-grounded in their discussion. Research limitations and future work are very well indicated and explained.
- References
The list of references is well prepared, the number of references is appropriate to the depth of the theme's approach in the article. The references are strong in the scope of this investigation. But these aspects must be corrected:
- Throughout the article, the authors indicate the references with name and year. And by the sequential numbering presented in the reference list, the article should be referenced only with numbers between square brackets;
- If authors prefer to remove numbering from the list of references and keep the referencing as it is, then, the term "et al." it should be put in italics, since in is a Latin term (not English).
Thank you for your thoughtful feedback regarding the referencing style used in our manuscript. Following your suggestion, we have adapted the referencing system within our manuscript. Now, instead of using the name and year, we have transitioned to referencing using numbers between square brackets in line with the sequential numbering presented in our reference list.
We have also ensured that the term "et al." is consistently italicized throughout the manuscript to acknowledge its Latin origin, as you rightly pointed out.
Reviewer 4 Report
1. The article lacks a clear purpose in the introduction. Although the goal appears in the summary, there is clearly no explanation of the reasons for taking up the topic with a clearly outlined goal. This element should absolutely be introduced. Considering the fact that in the theoretical part of the discussion goals are also formulated, as well as in the methodological part (e.g. "Our goal was to examine investors' reactions when their focal firms confront income-patible institutional pressures. and The main concern of the study was to compare the actual return of the listed firm's stock with the expected return after the release of the NSM list.") The set of goals should therefore be orderly. 2. The authors should, after arranging the objectives, describe what research approach they will use to prove each of the previously formulated hypotheses. The text shows that they will not assign methods to the hypotheses and will not verify the hypotheses, but they will pursue the research objective (one of many). It is necessary to organize the considerations, assigning one common hypothesis, show what method/methods it will be verified. In the same way, it is necessary to build detailed hypotheses and assign methods to them. There is a lack of a clearly formulated tree of goals, hypotheses and methods - the so-called road map. Such a map would show what is being studied and based on what approach. This element is not only important but essential to the introduction to the manuscript to understand the research study. 3. There is also no discussion as to why the chosen research method should be used and not others. This requires an explanation. 4. The least successful part is the ending. The conclusions are weak and insufficient, and there is no discussion of what the study contributed to the theory. There is no discussion as to whether the hypotheses that were previously put forward have been proven and how the study enriches the theory. The practical aspects of the study (where it can be used) are missing. There are no study limitations. At the end, there is no general conclusion in what direction research on the problem should be conducted in the future. The ending should be greatly expanded and revised.
Author Response
Reviewer 4:
Thank you for your appreciation of our work. Your detailed and constructive comments have greatly improved the paper. Below, we copy and respond to each of your comments in detail.
- The article lacks a clear purpose in the introduction. Although the goal appears in the summary, there is clearly no explanation of the reasons for taking up the topic with a clearly outlined goal. This element should absolutely be introduced. Considering the fact that in the theoretical part of the discussion goals are also formulated, as well as in the methodological part (e.g. "Our goal was to examine investors' reactions when their focal firms confront income-patible institutional pressures. and The main concern of the study was to compare the actual return of the listed firm's stock with the expected return after the release of the NSM list.") The set of goals should therefore be orderly.
Thank you for taking the time to review our manuscript and for your constructive comments. We appreciate your feedback regarding the clarity of our research objectives in the introduction.We have revised the introduction to more explicitly state the rationale for our research topic. We've also emphasized the importance of investor perceptions, particularly in the context of environmental management, and elaborated on the significance of our study given recent shifts in investor behavior and policy directions.Moreover, we have taken care to ensure that the goals of our research are consistently articulated throughout the manuscript. The revised introduction now clearly outlines our primary objective and the specific research questions that our study seeks to answer. Additionally, any repetitive or inconsistent statements of goals in the theoretical and methodological sections have been streamlined for clarity.
We hope that these revisions address your concerns and enhance the clarity and coherence of our manuscript. We are grateful for your insights, which have undoubtedly improved the quality of our work.
---------------------------Attached below for your reading convenience ---------------------
Introduction
The challenges of sustainable development are increasing, especially for manufacturing firms facing rigorous environmental management institutional pressures [1]. Despite the extensive body of research focused on how organizations address these challenges, there remains a gap in understanding the perceptions of external stakeholders—primarily investors—towards these pressures. Investors operate in a complex environment filled with uncertainty. In this backdrop, they continually strive to make sense of unforeseen events and determine how to respond to them[2].
The necessity to address this gap becomes imperative as understanding investor behavior can influence how firms position themselves in terms of sustainability practices. Consequently, this study emerges from a compelling need to discern how investors react when their invested organizations encounter conflicting institutional pressures, especially when characterized as polluting entities. Our inquiry gains significance as it not only aims to decipher investor responses but also endeavors to understand the weightage given to certain firm characteristics, such as political ties and green innovation capabilities, in shaping these responses.
With this in mind, the study sets forth with the following primary objectives:
To ascertain how investors interpret the polluting label of listed manufacturing firms.
To understand the perception of a firm's political ties and green innovative capabilities by investors, especially in the context of the polluting label.
To uncover how these political ties and innovation capabilities, when combined, mold investors' understanding of the polluting label.
To provide empirical weight to our objectives, we critically analyze the market value of listed manufacturing firms in China from 2010 to 2020. Our focus is on the National Specially-Monitored (NSM) program launched in 2007 to oversee heavy polluting firms. Given its nature, the NSM program serves as a distinctive quasi-experimental platform that facilitates causal inference. Moreover, with China's impending mandate for environmental data disclosure for its publicly listed firms, our exploration finds relevance by shedding light on the underlying reasons driving this transition towards greater environmental responsibility.
The role of investor perceptions in determining corporate actions cannot be under-stated. Recent studies have indicated that investors are increasingly considering environmental, social, and governance factors in their investment decisions, further underlining the importance of corporate environmental responsibility[3]. Moreover, in emerging markets such as China, where environmental degradation has been a significant concern, investors are uniquely positioned to drive corporate sustainability practices through their capital allocation decisions[4].
Furthermore, the association between a firm's political ties and its market valuation, particularly in the context of environmental controversies, has been the focus of recent investigations. Wu et al. (2018) noted that while political ties can offer protection and preferential resources in some contexts, they might also bind firms to governmental agendas and lead to entrenchment risks [5]. On the other hand, a firm's green innovation capabilities are emerging as a prominent factor in enhancing market value. Green innovation, as Huang and Li (2017) highlighted, can act as a significant differentiator in investor perceptions, offering firms a competitive advantage in a market that is increasingly prioritizing sustainable practices[6].
China's unique institutional environment, with its interplay between government, industry, and market forces, offers a fertile ground for such investigations. The Chinese government's rigorous push towards a more sustainable and green economy, as evidenced by its Five-Year Plans and other policy documents, further intensifies the need for a deeper understanding of investor behavior in the country's listed manufacturing sector [7].
With these perspectives, our study contributes to the growing body of literature that seeks to bridge the understanding of investor perceptions, corporate political ties, green innovation capabilities, and their combined influence on firm valuation in the face of environmental controversies.
-------------------------------------------------------------------------------------------------------
- The authors should, after arranging the objectives, describe what research approach they will use to prove each of the previously formulated hypotheses. The text shows that they will not assign methods to the hypotheses and will not verify the hypotheses, but they will pursue the research objective (one of many). It is necessary to organize the considerations, assigning one common hypothesis, show what method/methods it will be verified. In the same way, it is necessary to build detailed hypotheses and assign methods to them. There is a lack of a clearly formulated tree of goals, hypotheses and methods - the so-called road map. Such a map would show what is being studied and based on what approach. This element is not only important but essential to the introduction to the manuscript to understand the research study.
First and foremost, we appreciate your insightful feedback. In light of this, we have restructured Section 2 to incorporate a distinct “Road Map” segment to visually depict our research objectives, the corresponding hypotheses, and the methods we intend to employ for hypothesis validation.
---------------------------Attached below for your reading convenience ---------------------
2.1. Theoretical Roadmap: Goal, Hypotheses, and Methods
In this section, we present a structured approach detailing our research objectives, the corresponding hypotheses, and the methods we will employ to test each hypothesis:
Table 1 Roadmap of this study
|
Objective 1: Examine investors' perception of polluting labels and how this perception changes over time. |
|
- Hypothesis 1: The "polluting" label evokes negative perceptions among investors, but its impact lessens as time progresses. - Method: DID Analysis |
|
Objective 2: Understand the perception of a firm's political ties and green innovative capabilities by investors, especially in the context of the polluting label. |
|
- Hypothesis 2a: Ties with the central government positively influence investor perceptions. - Hypothesis 2b: Local political ties might be perceived negatively by investors. - Hypothesis 3: A strong green innovative capability is likely to foster positive investor perceptions. - Method: Two-stage approach |
|
Objective 3: Uncover how these political ties and innovation capabilities, when combined, mold investors' understanding of the polluting label. |
|
- Hypothesis 4a: The prominence of central ties positively moderates the negative impact of local ties on investors' perception. - Hypothesis 4b_1 and Hypothesis 4b_2: The influence of green innovative capability on perceptions of local ties. - Hypothesis 4c: The positive perception of innovative capability by investors is overshadowed by prominent central ties. - Method: Interaction effect test |
-------------------------------------------------------------------------------------------------------
- There is also no discussion as to why the chosen research method should be used and not others. This requires an explanation.
We appreciate your insightful feedback regarding the rationale for our methodological choices. In response to the concern about the lack of discussion regarding our chosen research methods, we have further elaborated on our rationale for adopting the PSM-DID and the two-stage models in the third section of Methodology.
---------------------------Attached below for your reading convenience ---------------------
- Given the NSM policy context, there's a requirement to compare firms labeled as polluters with those not labeled. The PSM-DID approach enables us to match these firms more effectively, ensuring they are comparable prior to the intervention, allowing for a more accurate estimation of the policy's effect on investor perceptions. PSM assists in reducing treatment selection bias, ensuring that the treated and control groups are balanced on observed pre-treatment covariates. The DID strategy aids in controlling for unobserved heterogeneities that remain constant over time, capturing the precise effect of the NSM policy on investor perceptions.
- Given that political ties and innovative capability might be correlated with other unobserved variables, this can introduce estimation bias. The two-stage approach allows us to consider all known control variables initially, and then in the second stage estimate our primary explanatory variables using the residuals from these controls. This procedure mitigates potential endogeneity issues. We developed a two-stage model to examine the effects of political ties and innovative capability on the investors’ reaction with all 1580 firms obtained by a 1:1 year by year propensity score matching (PSM), including 1670 NSM and 1670 non-NSM manufacturing firms. The stages were as follows:
-------------------------------------------------------------------------------------------------------
- The least successful part is the ending. The conclusions are weak and insufficient, and there is no discussion of what the study contributed to the theory. There is no discussion as to whether the hypotheses that were previously put forward have been proven and how the study enriches the theory. The practical aspects of the study (where it can be used) are missing. There are no study limitations. At the end, there is no general conclusion in what direction research on the problem should be conducted in the future. The ending should be greatly expanded and revised.
Thank you for your invaluable feedback regarding the conclusion of our paper. We recognize the importance of a robust conclusion that not only encapsulates the main findings but also elucidates our study's contributions to the prevailing theory. We understand your concerns about the absence of discussion surrounding the hypotheses and their implications. In our revised version, we will prioritize an in-depth discussion of how each hypothesis was addressed, drawing clear connections between our findings and their theoretical implications. Additionally, we will elaborate on the practical utility of our findings, ensuring that readers can comprehend the real-world implications of our study. We also acknowledge the oversight regarding the study's limitations. In our latest version, we detailed potential limitations and emphasize areas that could benefit from future research. Your insights have been crucial, and we're committed to significantly enhancing and expanding the conclusion to ensure a comprehensive understanding for our readers.
---------------------------Attached below for your reading convenience ---------------------
Discussion and Conclusion
In today's volatile business environments, organizations strive to showcase their unique standing to potential investors, emphasizing their unparalleled resource ad-vantages [50, 52, 53]. Building on this premise, the exploration of this study delves into the nuanced way investors interpret the 'polluting label', shedding light on how intertwined corporate resources and capabilities influence these perceptions.
Firstly, our study accentuates the pivotal role of the Prospect Theory in investor decision-making. As environments grow more intricate, investors, evolving into adept cue-seekers, begin to integrate the Prospect Theory. This allows them to meticulously analyze external uncertainties in tandem with internal corporate capabilities. Interestingly, this innovative application of the Prospect Theory to dissect the 'polluting label' unfurls a fresh paradigm in understanding investor behavior amidst uncertainties.
Furthermore, diverging from previous works that categorized political affiliations in a binary manner[20], this study provides a more textured understanding. By examining the strength and intricacies of these ties, our findings unravel the tug-of-war businesses often confront between central and local government affiliations. Such an approach augments our comprehension of the myriad ways these political linkages mold investor perceptions.
Transitioning to the realm of innovative capabilities, it's evident that, notwithstanding the challenges ushered in by the 'polluting label', robust corporate innovative prowess still resonates positively with investors. However, this favorable perception tends to diminish when juxtaposed with dominant central ties, echoing the timeless principle that inherent corporate vigor invariably garners investor appreciation.
On the practical front, firms grappling with the repercussions of a 'polluting label' can glean valuable insights. Strengthening central political ties and bolstering innovative capabilities can serve as twin pillars to counterbalance and potentially recalibrate negative investor perceptions.
Yet, as with any comprehensive study, certain boundaries and constraints delimit our research. Our empirical focus on the 'polluting label' primarily catered to scenarios with a stark demarcation between dominant and subordinate logics. Consequently, the extrapolation of our conclusions to other contexts mandates prudence and circumspection. Moreover, while our narrative predominantly revolves around political ties and innovative capabilities vis-à-vis the 'polluting label', the canvas remains expansive. Unraveling other potential organizational cues, like corporate reputation, might offer invaluable in-sights, especially when businesses find themselves at crossroads with conflicting institutional mandates.
In conclusion, our endeavor bridges the intricate interplay between investor perceptions, political dynamics, and corporate innovation within the 'polluting label' context. As sustainability emerges as the clarion call of modern businesses, comprehending these intertwined dynamics assumes paramount importance.
-------------------------------------------------------------------------------------------------------

Reviewer 5 Report
Dear author,
Thank you for sending your paper to the journal.
The paper needs substantial revision in the following sections:
1-It is recommended to conduct proofreading on the paper.
2-Line 91 should be fully revised
2-The originality of the paper is not illustrated in the introduction, so you need to improve this part.
3- The theoretical issues should be improved with the most recent studies; some are here:
-The impact of political ties on firms’ innovation capability: Evidence from China
-Knowledge sharing barriers and knowledge sharing facilitators in innovation
-The Effect of CO2 Gas Emissions on the Market Value, Price and Shares Returns
-The impact of sustainability‐oriented dynamic capabilities on firm growth: Investigating the green supply chain management and green political capabilities
4-It is recommended to date the data (up to 2017) to 2022.
5-The paper needs to conduct additional analyses and robustness test
6- The research implications should be stated according to each hypothesis's results.
Needs a minor revision
Author Response
Reviewer 5:
Thank you for your appreciation of our work. Your detailed and constructive comments have greatly improved the paper. Below, we copy and respond to each of your comments in detail.
The paper needs substantial revision in the following sections:
1-It is recommended to conduct proofreading on the paper.
Thank you for your recommendation. We acknowledge the importance of delivering a well-polished manuscript, free from grammatical and stylistic errors. We promptly engaged in a comprehensive proofreading process to ensure the paper meets the highest standards of clarity and precision.
- Line 91 should be fully revised
Thank you for pointing out the issue with Line 91. Taking your feedback into account, along with the suggestions from other reviewers, we have revised the entire paragraph for clarity and coherence. We appreciate your meticulous attention to detail and guidance.
- The originality of the paper is not illustrated in the introduction, so you need to improve this part.
Thank you for the valuable feedback on emphasizing the originality of our paper in the introduction. We have now emphasized the unique perspectives and gaps our study addresses, contrasting our research approach and focus with existing literature. This revised introduction better showcases the novelty of our study, underscoring its contribution to the academic discourse on sustainable development and investor behavior. We hope this amendment meets the expectations and further clarifies the significance of our research.
---------------------------Attached below for your reading convenience ---------------------
Introduction
The challenges of sustainable development are increasing, especially for manufacturing firms facing rigorous environmental management institutional pressures [1]. Despite the extensive body of research focused on how organizations address these challenges, there remains a gap in understanding the perceptions of external stakeholders—primarily investors—towards these pressures. Investors operate in a complex environment filled with uncertainty. In this backdrop, they continually strive to make sense of unforeseen events and determine how to respond to them[2].
What sets our research apart is its exploration into the interplay between investor perceptions and organizational characteristics, a dimension less traversed in literature. Understanding investor behavior becomes pivotal, as it holds the potential to redefine how firms prioritize and implement sustainability practices. This study emanates from a pressing need to decipher investor reactions when their invested firms confront the duality of institutional pressures, particularly when labeled as polluting entities. This research is distinguished by its commitment to not only decode investor reactions but also probe the influence of certain firm traits, such as political affiliations and eco-innovation competencies, in sculpting these reactions. With this in mind, the study sets forth with the fol-lowing primary objectives:
First, to ascertain how investors interpret the polluting label of listed manufacturing firms.
Second, to understand the perception of a firm's political ties and green innovative capabilities by investors, especially in the context of the polluting label.
Third, to uncover how these political ties and innovation capabilities, when combined, mold investors' understanding of the polluting label.
To provide empirical weight to our objectives, we critically analyze the market value of listed manufacturing firms in China from 2010 to 2020. Our focus is on the National Specially-Monitored (NSM) program launched in 2007 to oversee heavy polluting firms. Given its nature, the NSM program serves as a distinctive quasi-experimental platform that facilitates causal inference. Moreover, with China's impending mandate for environmental data disclosure for its publicly listed firms, our exploration finds relevance by shedding light on the underlying reasons driving this transition towards greater environmental responsibility.
The role of investor perceptions in determining corporate actions cannot be under-stated. Recent studies have indicated that investors are increasingly considering environmental, social, and governance factors in their investment decisions, further underlining the importance of corporate environmental responsibility[3]. Moreover, in emerging markets such as China, where environmental degradation has been a significant concern, investors are uniquely positioned to drive corporate sustainability practices through their capital allocation decisions[4].
Furthermore, the association between a firm's political ties and its market valuation, particularly in the context of environmental controversies, has been the focus of recent investigations. Wu et al. (2018) noted that while political ties can offer protection and preferential resources in some contexts, they might also bind firms to governmental agendas and lead to entrenchment risks [5]. On the other hand, a firm's green innovation capabilities are emerging as a prominent factor in enhancing market value. Green innovation, as Huang and Li (2017) highlighted, can act as a significant differentiator in investor perceptions, offering firms a competitive advantage in a market that is increasingly prioritizing sustainable practices[6].
China's unique institutional environment, with its interplay between government, industry, and market forces, offers a fertile ground for such investigations. The Chinese government's rigorous push towards a more sustainable and green economy, as evidenced by its Five-Year Plans and other policy documents, further intensifies the need for a deeper understanding of investor behavior in the country's listed manufacturing sector [7].
With these perspectives, this study contributes to the growing body of literature that seeks to bridge the understanding of investor perceptions, corporate political ties, green innovation capabilities, and their combined influence on firm valuation in the face of environmental controversies.
-------------------------------------------------------------------------------------------------------
4- The theoretical issues should be improved with the most recent studies; some are here:
-[1]The impact of political ties on firms’ innovation capability: Evidence from China
-[2]Knowledge sharing barriers and knowledge sharing facilitators in innovation
-[3]The Effect of CO2 Gas Emissions on the Market Value, Price and Shares Returns
-[4]The impact of sustainability‐oriented dynamic capabilities on firm growth: Investigating the green supply chain management and green political capabilities
Thank you for highlighting the recent studies that could further bolster the theoretical foundation of our paper. We appreciate the emphasis on ensuring our research remains relevant and up-to-date. In response to your feedback:
We have delved into the study on "The impact of political ties on firms’ innovation capability: Evidence from China" to better integrate the nuances of how political ties influence innovation, particularly within the Chinese context (in the 2.3 section).
The article "Knowledge sharing barriers and knowledge sharing facilitators in innovation" has been instrumental in providing a more comprehensive perspective on the factors that either hinder or promote innovation. It has been used to explain the relationship between political ties and innovative capability (in the 2.4 section).
We acknowledge the importance of understanding the direct and indirect effects of CO2 gas emissions on market dynamics. By referring to "The Effect of CO2 Gas Emissions on the Market Value, Price, and Shares Returns", we offered a more robust correlation between sustainability practices, emissions, and market value (in the 2.2 section).
Lastly, the research on "The impact of sustainability‐oriented dynamic capabilities on firm growth: Investigating the green supply chain management and green political capabilities" has be integrated to enrich our discussion on how sustainability-focused capabilities influence firm growth (in the 2.3 section).
We are committed to revising our paper to incorporate the insights from these studies, ensuring a more robust theoretical framework. Once again, thank you for your insightful feedback.
---------------------------Attached below for your reading convenience ---------------------
2.2. Investor’s perception of “Polluting” label
Investors evaluate a firm's environmental commitment and overall quality when confronted with a "polluting" label, assessing its corporate resources, capabilities, and potential for sustainable growth. This evaluation fundamentally hinges on the Prospect Theory [8], which postulates that individuals assess decisions predicated on anticipated gains or losses.
The nature of these gains and losses, and the resulting risk attitudes, have been the focus of extensive research. For instance, Van Dijk and Van Knippenberg (1996) found that heightened uncertainty can exacerbate loss aversion [9]. This is further corroborated by Shogren et al. (1994), who illustrated the intensified sensitivity towards irreplaceable losses [10]. Additionally, the influence of acquisition difficulty on the judgment of gains and losses has been underlined by Loewenstein & Issacharoff (1994)[11].
The ever-evolving discourse on sustainability and corporate responsibility has deepened the complexities surrounding the "polluting" label. Firms that appear on lists such as the NSM often send out signals of prioritizing immediate profits over sustainable practices. This, in turn, places them under the scrutiny of both the public and regulatory bodies. In the face of these signals, it's plausible to assume that investors' loss perceptions are sharpened, given the potential reputational and financial implications for the firm.
In a modern context, research has emphasized the financial implications of environmental disregard. For instance, a study highlighted that carbon-intensive companies, or those flagged for high CO2 emissions, often experience a stock market penalty, reinforcing the financial repercussions of the "polluting" label [12, 13]. Another notable study has elucidated that, increasingly, investors, especially institutional ones, are integrating Environmental, Social, and Governance (ESG) criteria into their investment decisions [14]. This implies that labels like "polluting" can indeed have profound impacts on investors' perceptions and decision-making processes.
However, Prospect Theory posits that individuals might demonstrate heightened risk tolerance in the face of negative outcomes. Given this, it is conceivable that investors might retain holdings in firms with a long-standing presence on the NSM list, banking on the potential for future rebounds or strategic shifts towards sustainability. Over time, this might attenuate the initially potent negativity associated with the "polluting" label.
Hypothesis 1 (H1): The "polluting" label evokes negative perceptions among investors, but its impact lessens as time progresses.
2.3. Roles of Political Ties and Green Innovative Capability on Investors’ Perception
Political Ties
Central Government Ties: Firms' strategic relationships with central governments can confer considerable advantages, and this has been extensively documented in academic literature [15]. The nature and potential of these ties traverse beyond mere information access and can profoundly shape a firm's trajectory. An empirical examination from China illuminates how such political affiliations considerably bolster a firm’s innovation capabilities [16]. This sheds light on the multifaceted nature of these ties, suggesting they also act as catalysts that spur innovation by providing firms with advantageous resources and support.
These benefits of central government ties also encompass enabling firms to gain a deeper understanding of policy design, anticipate policy shifts, and adeptly navigate the implications of those shifts [17, 18]. Such foresight and adaptability are invaluable, especially given the dynamic and often unpredictable nature of the political landscape.
Firms with central government ties also enjoy reduced uncertainty, especially in environments characterized by volatile regulatory frameworks[19]. By tapping into these relationships, firms access privileged insights, positioning them in a proactive rather than a reactive stance[20]. Yet, while these ties confer numerous advantages, they also entail specific responsibilities and expectations. Companies that deviate from central government guidelines, or fail to capitalize on their relationships effectively, can face severe repercussions, ranging from reduced support to punitive actions [21, 22].
Incorporating the above understanding, it becomes evident that the presence and depth of ties with the central government can considerably sway investor perceptions. Investors, recognizing the multifaceted advantages of such ties, are likely to view them as indicators of a firm's strategic positioning and potential resilience against policy shocks.
Hypothesis 2a (H2a): Ties with the central government positively influence investor perceptions.
Local political ties, epitomized by affiliations at sub-national levels such as provinces, can serve as a double-edged sword for firms [23]. On the one hand, these ties can be instrumental in reducing regulatory uncertainty, facilitating smoother access to local resources, and potentially aiding in swift navigation of regional bureaucracies [24]. Furthermore, they can engender a closer alignment with local stakeholders, boosting a firm's social capital and reputation within the community [25].
However, challenges arise when there's a misalignment between local and central objectives. Such divergences can place firms in precarious situations. While local ties might help navigate regional challenges, they might simultaneously complicate a firm's relationship with central authorities, especially when local agendas clash with broader national mandates. Firms heavily embedded in local politics might also face challenges if regional leadership undergoes significant change or if there's a major policy shift at the local level [26]. Consequently, the potential risks associated with these ties could overshadow their benefits, especially in the eyes of investors who are keenly aware of the balancing act firms must perform between local affiliations and central directives [27, 28].
Given the nuanced interplay of benefits and potential detriments associated with local political ties, investors' perceptions might lean towards caution. This cautious outlook stems from the inherent risks these ties introduce, especially when considering the overarching objectives of programs like NSM and their potential repercussions.
Hypothesis 2b (H2b): Local political ties might be perceived negatively by investors.
Green Innovative Capability
The realm of green innovation has been receiving increased attention as both an environmental imperative and a business opportunity. Delving deeper than the earlier observation of the importance of sustainability-oriented dynamic capabilities, recent research underscores its role in positioning firms strategically in an eco-conscious market [29]. Such sustainable strategies, as corroborated by Hart and Dowell (2011), can provide dual benefits – they not only cater to growing global environmental concerns but also drive a competitive edge in the market [30].
A key dimension of green innovation is its capacity to synergize economic performance with environmental sustainability. This encompasses the development of new products, practices, and organizational processes that simultaneously boost economic returns and reduce ecological footprints[31]. In the evolving landscape, green innovation isn't just an add-on; it's becoming integral for companies aiming for long-term success. Notably, firms with a "polluting" label stand to gain considerably by demonstrating a robust commitment to green innovation, as this can potentially offset negative perceptions and highlight a proactive approach to sustainability [32].
Moreover, Chen et al. (2019) emphasized
the importance of integrating green innovation strategies within core business models. They argued that such integration not only helps firms meet stringent regulatory standards but also positions them favorably among environmentally-conscious investors[33]. Thus, in the backdrop of these insights, and considering the criticality of green innovative capabilities for labeled polluters, the following hypothesis is presented:
Hypothesis 3 (H3): A strong green innovative capability is likely to foster positive investor perceptions.
2.4. Interplay between Political Ties and Innovative Capability in Influencing Investors’ Perception
Central and Local Political Ties Interaction
Firms often navigate the political landscape by fostering relationships at both the central and local levels. While the intertwined nature of these relationships offers a layered understanding, the coexistence may sometimes lead to investor apprehension [34]. A firm's affiliation with the central government tends to offer advanced access to resources, policy insights, and a reduced risk of punitive actions [17]. Drawing upon the certainty effect embedded in the Prospect Theory, investors are likely to gravitate towards the tangible benefits emanating from central political ties, potentially overshadowing the intricacies and perceived risks associated with local affiliations.
Hypothesis 4a (H4a): Central political ties serve as a buffer, attenuating the potential adverse impact of local ties on investors' perception.
Local Ties and Innovative Capability Interaction
Recent studies have illuminated the intricacies of innovation within the context of knowledge sharing and transfer, highlighting both potential roadblocks and catalysts [35]. When local political ties enter the equation, they can either bolster or hinder a firm's innovative pursuits, contingent on the nature of these relationships. With the continuing debate on whether relationships and capabilities offer alternative or complementary value propositions[36], understanding the investor perception becomes crucial. Drawing from both the Prospect Theory and the Portfolio Selection Theory[37], the intricate balance of green innovation's inherent risks and the uncertainties underpinned by dominant local ties can result in differential investor sentiments.
Hypothesis 4b_1 (H4b_1): A robust green innovative capability acts as a counterbalance, offsetting the negative investor perceptions tied to local political affiliations.
Hypothesis 4b_2 (H4b_2): Intense green innovative capability, when juxtaposed with prominent local ties, exacerbates investor concerns due to compounded uncertainties.
Central Ties and Innovative Capability Interaction
For firms deeply entrenched with the central government, the interplay between political ties and innovative capabilities offers a unique dimension. While innovation can herald adaptability, growth, and potential label mitigation, the journey is riddled with uncertainties [38, 39]. In contrast, central political affiliations can serve as a beacon of stability, often translating to economic rewards and competitive edge[40, 41]. Drawing from the irreplaceability principle of Shogren et al. (1994) and the recent findings by Wang et al. (2021), which indicate that investors often prefer known certainties over unknown risks [10, 42]. it is posited that robust central affiliations might eclipse the perceived benefits of innovation.
Hypothesis 4c (H4c): The positive perception of innovative capability by investors is overshadowed by prominent central ties.
-------------------------------------------------------------------------------------------------------
- It is recommended to date the data (up to 2017) to 2022.
Thank you for your suggestion to update our data to include the years up to 2022. I'd like to provide some context regarding our data selection:
The NSM (National Specially-Monitored) program undergoes updates either at the beginning of each year or at the end of the previous year. Due to factors, potentially including the pandemic, starting from 2021, not all provinces have disclosed their list of nationally monitored enterprises. This inconsistency makes it challenging to construct a comprehensive dataset that represents the entirety of China.
Moreover, our choice to begin from 2010 is deliberate. We aim to observe the long-term impact of the NSM list on enterprises. A decade, in our opinion, offers an appropriate timeframe to study such effects thoroughly.
We appreciate your understanding and will consider your feedback in our future research endeavors.
- The paper needs to conduct additional analyses and robustness test
Thank you for emphasizing the importance of additional analyses and robustness tests. In response to your feedback, we have incorporated a robustness check section in the paper to ensure the validity and reliability of our findings. We believe that this addition will strengthen the paper's overall rigor and trustworthiness. We appreciate your valuable insights and hope this revision meets your expectations.
---------------------------Attached below for your reading convenience ---------------------
- Robustness test
To ensure the reliability of our primary findings that suggest a negative association between the polluting label and investors' perception, we executed several robustness checks. Initially employing a Difference-in-Differences (DID) approach, we assessed the impact of polluting labels on investor perception and discovered a significant decline in the value of companies listed on the NSM list. After applying a 1:1 propensity score matching (PSM), we derived a comprehensive sample. Subsequently, our treatment variable, which identifies whether a company is on the NSM list or not, was tested in a two-stage model, revealing a notably negative impact. This outcome further substantiated the adverse effect of the polluting label on firms. To deepen our scrutiny, we conducted a mixed cross-sectional regression analysis of all NSM companies from 2016 to 2020. Consistent with our primary results, this method illustrated a positive correlation between central political ties and investors' perception (BHAR) with β=0.244 (p<0.01) and a similarly positive relationship for innovative capability with β=0.153 (p<0.01). However, local political ties displayed a negative association with investors' perception, as indicated by β=-0.102 (p<0.10). Delving deeper, we found positive effects on investors’ perception of the polluting label from both the interaction between central and local ties (β=0.176, p<0.05) and the interaction between local ties and innovative capability (β=0.198, p<0.05). Yet, the interaction between central ties and innovative capability showcased a negative influence with β=-0.260 (p<0.01). These robustness checks, taken collectively, corroborate the stability and consistency of our initial findings.
In conclusion, our rigorous robustness tests consistently validate the initial findings. The polluting label bears a negative association with investors' perception, substantiating the sentiment that such labels pose a tangible detriment to firms. However, the strength and direction of this relationship are notably influenced by political ties and innovative capabilities. These insights not only underscore the importance of considering institutional relationships and internal firm competencies when interpreting investor behavior but also emphasize the multifaceted impact of external labels on firm valuation in the eyes of investors.
-------------------------------------------------------------------------------------------------------
- The research implications should be stated according to each hypothesis's results.
We appreciate your emphasis on clarifying the research implications vis-à-vis each hypothesis. You're absolutely right; it's pivotal to delineate the implications of each hypothesis's results to provide readers with a nuanced understanding of our study's outcomes. In the revised manuscript, we have incorporated a dedicated section that correlates each hypothesis with its findings, ensuring a thorough articulation of the research implications. This will not only enhance the clarity of our paper but will also allow our audience to gauge the significance of each result within the broader research context. Your feedback is instrumental in refining our work, and we're grateful for your insights.
---------------------------Attached below for your reading convenience ---------------------
Discussion and Conclusion
In today's volatile business environments, organizations strive to showcase their unique standing to potential investors, emphasizing their unparalleled resource advantages [50, 52, 53]. Building on this premise, the exploration of this study delves into the nuanced way investors interpret the 'polluting label', shedding light on how intertwined corporate resources and capabilities influence these perceptions.
Firstly, our study accentuates the pivotal role of the Prospect Theory in investor decision-making. As environments grow more intricate, investors, evolving into adept cue-seekers, begin to integrate the Prospect Theory. This allows them to meticulously analyze external uncertainties in tandem with internal corporate capabilities. Interestingly, this innovative application of the Prospect Theory to dissect the 'polluting label' unfurls a fresh paradigm in understanding investor behavior amidst uncertainties.
Furthermore, diverging from previous works that categorized political affiliations in a binary manner[20], this study provides a more textured understanding. By examining the strength and intricacies of these ties, our findings unravel the tug-of-war businesses often confront between central and local government affiliations. Such an approach augments our comprehension of the myriad ways these political linkages mold investor perceptions.
Transitioning to the realm of innovative capabilities, it's evident that, notwithstanding the challenges ushered in by the 'polluting label', robust corporate innovative prowess still resonates positively with investors. However, this favorable perception tends to diminish when juxtaposed with dominant central ties, echoing the timeless principle that inherent corporate vigor invariably garners investor appreciation.
On the practical front, firms grappling with the repercussions of a 'polluting label' can glean valuable insights. Strengthening central political ties and bolstering innovative capabilities can serve as twin pillars to counterbalance and potentially recalibrate negative investor perceptions.
Yet, as with any comprehensive study, certain boundaries and constraints delimit our research. Our empirical focus on the 'polluting label' primarily catered to scenarios with a stark demarcation between dominant and subordinate logics. Consequently, the extrapolation of our conclusions to other contexts mandates prudence and circumspection. Moreover, while our narrative predominantly revolves around political ties and innovative capabilities vis-à-vis the 'polluting label', the canvas remains expansive. Unraveling other potential organizational cues, like corporate reputation, might offer invaluable insights, especially when businesses find themselves at crossroads with conflicting institutional mandates.
In conclusion, our endeavor bridges the intricate interplay between investor perceptions, political dynamics, and corporate innovation within the 'polluting label' context. As sustainability emerges as the clarion call of modern businesses, comprehending these intertwined dynamics assumes paramount importance.
-------------------------------------------------------------------------------------------------------

Round 2
Reviewer 1 Report
The authors did not revise the manuscript in accordance with my review comments, including comments 1, 2, 3, 5, 6, and 7. My decision is rejected.
Good luck for your work!
Moderate editing of the English language is required.
Author Response
Dear Reviewer,
I hope this message finds you well. Firstly, I'd like to express my appreciation for the time and effort you've invested in reviewing my manuscript.
I've noted your feedback from the second round of reviews, indicating that certain comments might not have been adequately addressed in the revised manuscript. I apologize for any oversight and wish to provide a detailed account of how each of your comments was considered and incorporated.
First, your first and third comments concerning the article's structure and theoretical framework, and another Reviewer recommended that we should elucidate our research objectives, hypotheses, and methods through a clear roadmap. As a result, both the introduction and the subsequent section, which covers the literature review and hypothesis development, have been revised. The conclusion of the introduction now presents our three main research objectives for this study. Moving on to the second section, we introduced a new subsection (2.1) dedicated to a paper roadmap. This roadmap systematically showcases, for each research objective, the associated hypotheses we seek to examine and the respective methodologies employed for validation. Subsections 2.2 to 2.4 further delve into hypothesis development, anchored in pertinent literature. We are confident that these refinements effectively convey the article's structural and theoretical coherence.
Second, in response to Comment 2, you suggested that we should incorporate the most recent literature to bolster the research hypothesis. Notably, another Reviewer had earlier provided four pertinent studies during the review process. We complemented these with several of our own selections. To ensure clarity and coherence in the second section, we undertook a comprehensive rewrite. The key references we have now included are:
- Salehi, M., et al., The effect of CO2 gas emissions on the market value, price and shares returns. Energies, 2022. 15(23): p. 9221.
- Xu, L., et al., Spillover effects and nonlinear correlations between carbon emissions and stock markets: An empirical analysis of China's carbon-intensive industries. Energy Economics, 2022. 111: p. 106071.
- Wang, K., et al., The impact of political ties on firms’ innovation capability: Evidence from China. Asia Pacific Journal of Management, 2023: p. 1-33.
- Yi, Y. and P. Demirel, The impact of sustainability‐oriented dynamic capabilities on firm growth: Investigating the green supply chain management and green political capabilities. Business Strategy and the Environment, 2023.
- Salehi, M. and S.A. Sadeq Alanbari, Knowledge sharing barriers and knowledge sharing facilitators in innovation. European Journal of Innovation Management, 2023.
- Wang, T., T. Zhang, and Z. Shou, The double-edged sword effect of political ties on performance in emerging markets: The mediation of innovation capability and legitimacy. Asia Pacific Journal of Management, 2021. 38: p. 1003-1030.
In response to your Comment 5, which inquired about our usage of 'Treated' and 'Time' in relation to the "difference-in-difference causal inferences" mentioned in our abstract, we have made revisions. Specifically, our revised abstract now clarifies: "...firms under the National Specially-Monitored (NSM) program are designated as the treated group, while those not in the NSM program serve as the control group. The 'Time' variable denotes the period following the NSM program's initiation..."
Addressing your Comment 6 on the "Results" section: While we primarily reported our findings, the reviewer suggested enhancing the section by integrating insights from 2 or 3 seminal studies in the field for added context. To address this, we have incorporated references from relevant studies to compare and contextualize our results. We provide a comprehensive breakdown and the included references in the attached response letter for more detailed insight.
Regarding your Comment 7 about the clarity of the "Discussion" and "Conclusion" sections: Feedback highlighted the need for restructuring around four core elements: main findings, theoretical contributions, managerial implications, and limitations with directions for future research. To address this:
(1) In the opening paragraph, we clearly define our research question.
(2) The subsequent three paragraphs focus on our theoretical contributions, grounded in our primary research outcomes.
(3) The fifth paragraph elaborates on the practical managerial implications derived from our findings.
(4) In the sixth paragraph, we discuss the study's limitations and suggest potential areas for future research.
(5) Finally, the concluding paragraph offers a concise summary of our research.
I hope this clarifies how I've approached your feedback in the revision. If there are any areas that still require adjustment or further elaboration, I am more than willing to make the necessary changes to enhance the manuscript's quality.
Thank you once again for your invaluable insights, and I sincerely hope the above explanations resonate with your expectations.
Warm regards,
Authors

Reviewer 2 Report
(1)In Model (5), why are the coefficients of D-1, D-2, …D7 same as β15?
(2)The DID model needs to have the parallel trend test, however, this test is not shown in the manuscript.
(3)Authors added robustness tests into the paper, but the robust regressions’ results are not shown in the paper.
(4) The paper should focus the change effect between after and before a listed company is labeled as the polluting firm, and external political ties and internal green innovation capabilities are only the moderation effects.
Moderate editing of English language required
Author Response
Dear reviewer:
At the outset, we want to thank you for the detailed and constructive comments, which have provided much food for thoughts and great opportunities for us to improve the theorizing and empiric of the paper. Below, we copy and respond to each of your comments in detail.
(1)In Model (5), why are the coefficients of D-1, D-2, …D7 same as β15?
We sincerely apologize for the oversight in our equations. The necessary corrections have been made, and we appreciate your attention to this detail. Building on the approach of Beck et al. (2010)[52], this study adopts a dynamic model to examine the influence of the 'polluting label' on investor perceptions. We consider a time span of 15 years, encompassing 5 years before the acquisition of the 'polluting label' and 10 years subsequent to its assignment.
---------------------------Attached below for your reading convenience ---------------------
-------------------------------------------------------------------------------------------------------
(2)The DID model needs to have the parallel trend test, however, this test is not shown in the manuscript.
First and foremost, we sincerely apologize for not clarifying this matter adequately in the original manuscript. Our paper employs a multi-period DID model, wherein the treatment mechanism occurs with time disparities. Different enterprises are marked with the 'polluting label' at different time points. The parallel trends test in this context deviates from the traditional DID. We have drawn upon the methods of Beck et al.(2010), considering a 15-year window period (from 5 years before receiving the 'polluting label' to 10 years after). We then utilized both graphical and regression methods to examine the policy's parallel trends and dynamic effects, which are discussed in our results.
---------------------------Attached below for your reading convenience ---------------------
The prerequisite for using the DID (Difference-in-Differences) method is to satisfy the parallel trends assumption. Following the approach of Beck et al. (2010), this paper utilizes both graphical and regression methods to examine the policy's parallel trends and dynamic effects.
Firstly, the graphical method is employed to compare the trends in enterprise value before and after receiving the 'polluting label'. As depicted in Figure 1, there was no significant difference in urban innovation levels between the treatment and control groups before the 'polluting label' was assigned, confirming the parallel trends. It is evident from Figure 1 that after enterprises received the 'polluting label', the policy effect emerged and strengthened over time. Furthermore, although 7 years after the impact of NSM the market value rebounded, the coefficients on to were not significant at the 5% level. Therefore, this study excluded these samples that had been continuously monitored for more than 7 years, and re-tested them. Figure 2 indicates that the results are robust.
Subsequently, to further validate the parallel trends and assess the dynamic policy effects on NSM, regression results are presented in Table 4. Table 4 indicates that, after being labeled as 'polluting', the policy coefficient is significantly different from 0, while before receiving the label, the policy coefficient is not significantly different from 0. This further demonstrates that the treatment and control city groups satisfied the parallel trends assumption before policy implementation. Additionally, after policy enactment, the absolute value of the regression coefficient increases annually. Consequently, the 'polluting label' has a long-term and stable impact on enterprise value. Hypothesis 1 is thus validated."
|
|
|
*The figure plots the impact of NSM duration on the market value of NSM firms. A 15-year window, spanning from 5 years before NSM until 9 years on the list. The dashed lines represent 95% confidence intervals. Figure 1. The dynamic impact of NSM program on the market value. |
|
|
|
*The figure plots the impact of NSM duration on the market value of NSM firms. A 12-year window, spanning from 5 years before NSM until 7 years on the list. The dashed lines represent 95% confidence intervals. Figure 2. The dynamic impact of NSM on the market value. |
Table 4. Descriptive statistics and bivariate correlations.
|
|
Coefficient |
|
Coefficient |
|
Before5 |
0.002 (0.166) |
After3 |
0.179 (0.055)*** |
|
Before4 |
0.010 (0.079) |
After4 |
0.182 (0.072)** |
|
Before3 |
-0.008 (0.159) |
After5 |
0.264 (0.074)*** |
|
Before2 |
0.006 (0.052) |
After6 |
0.328 (0.081)*** |
|
Before1 |
0.004 (0.070) |
After7 |
0.415 (0.105)*** |
|
Current |
- |
After8 |
0.266 (0.166) |
|
After1 |
0.112 (0.053)** |
After9 |
0.201 (0.174) |
|
After2 |
0.191 (0.052)*** |
|
|
|
Constant |
-0.936 (0.268)*** |
||
|
Obs. |
3340 |
||
|
R2 |
0.289 |
||
-------------------------------------------------------------------------------------------------------
(3) Authors added robustness tests into the paper, but the robust regressions’ results are not shown in the paper.
We sincerely apologize. In the previous version, in an effort to conserve space in the manuscript, we directly described some regression result coefficients in the robustness test section without presenting them in a table. We have now included a complete table of regression results.
---------------------------Attached below for your reading convenience ---------------------
Table 8. The regression results of the robustness test
|
Variables |
Model 1 |
Model 2 |
Model 3 |
Model 4 |
Model 5 |
|
Intercept |
0.087 (0.022)*** |
0.088 (0.022)*** |
0.085 (0.022)*** |
0.087 (0.022)*** |
0.081 (0.023)*** |
|
Central tie |
0.213 (0.065)*** |
0.221 (0.064)*** |
0.218 (0.065)*** |
0.214 (0.064)*** |
0.244 (0.074)*** |
|
Local tie |
-0.098 (0.051)* |
-0.097 (0.051)* |
-0.116 (0.052)* |
-0.098 (0.051)* |
-0.102 (0.052)* |
|
Green innovative capability |
0.144 (0.035)*** |
0.147 (0.039)*** |
0.145 (0.033)*** |
0.171 (0.051)*** |
0.153 (0.033)*** |
|
Central tie* Local tie |
|
0.222 (0.088)** |
|
|
0.176 (0.071)** |
|
Green innovative capability*Local tie |
|
|
|
0.196 (0.074)** |
0.198 (0.088)** |
|
Central tie * Green innovative capability |
|
|
-0.299 (0.053)*** |
|
-0.260 (0.043)*** |
|
Type of actual controller (dummy) |
Included |
Included |
Included |
Included |
Included |
|
Year (dummy) |
Included |
Included |
Included |
Included |
Included |
|
Model F-value |
18.70*** |
18.88*** |
18.12*** |
12.14*** |
12.44*** |
|
R² |
0.341 |
0.334 |
0.331 |
0.345 |
0.369 |
|
Adjusted R² |
0.331 |
0.323 |
0.322 |
0.335 |
0.362 |
-------------------------------------------------------------------------------------------------------
(4) The paper should focus the change effect between after and before a listed company is labeled as the polluting firm, and external political ties and internal green innovation capabilities are only the moderation effects.
We wholeheartedly agree with your perspective. The primary research question of the paper revolves around the impact of the 'polluting label' on investors’ perception. Our principal finding underscores that companies, once labeled, witness a pronounced decline in their value. The exploration of the moderating roles of political affiliations and green innovation is motivated by two key considerations:
First, the 'polluting label' discernibly diminishes a company's value.
Second, given the inherent nature of the labeling process, it presents a formidable challenge for firms to disassociate from the 'polluting label'. As per the stipulations of the NMF program, companies whose pollutant emissions surpass the defined threshold are invariably branded as 'polluting'. This places firms in a predominantly reactive stance. Despite adopting mitigative measures, the removal of the 'polluting label' remains a challenge, more so for larger enterprises.
In light of the pronounced adverse effects of the 'polluting label' and its persistent nature, an imperative question emerges: How might firms navigate and mitigate the ramifications of such a designation, especially in an environment where countless entities face a similar predicament? Accordingly, this study delves into both internal and external perspectives, investigating the potential moderating roles of political ties and firm-level innovation. We anticipate that these insights will provide valuable guidance for corporate strategic planning.

Reviewer 4 Report
The article has been corrected. The manuscript may be published.
Author Response
Your insights and feedback are invaluable to ensuring the quality and integrity of the work. I appreciate the time and effort you are investing in this review process.
Reviewer 5 Report
Dear authors;
You well done the current version and incorporated my comments on the current version.
Author Response

(The authors gave the same response as above.)

Round 3
Reviewer 2 Report
Authors have not solve perfectly the problems that I had raiesed.
Moderate editing of English language required